# Safe Online Learning via Smooth Safety-Structured Policy Composition

## Abstract

Safe online reinforcement learning requires policies to respect safety constraints while maintaining smooth optimization dynamics. Existing approaches typically rely on either strict safety enforcement via action interventions, which introduce discontinuities in system interaction and learning, or soft safety constraint formulations, which preserve smooth learning but provide limited safety assurance. We propose *AutoSafe*, a safety-aware policy architecture that integrates structured safety monitoring and intervention directly into the action generation process. This design enables smooth, risk-dependent transitions between performance-driven and safety-preserving behaviors, resulting in continuous online interaction and learning dynamics. Empirical results across a suite of continuous-control benchmarks demonstrate strong safety enforcement without sacrificing learning smoothness. We further validate AutoSafe on a physical cart-pole system, highlighting its practical effectiveness for safe online learning in the real world.

## 1 Introduction

Enabling deep reinforcement learning (DRL) agents to interact safely with their environment while maintaining stable and continual policy improvement remains a central challenge for real-world learning systems (Ibarz et al., 2021). In safety-critical applications, however, safety requirements must be enforced as hard constraints, significantly complicating online learning.

Among existing safe reinforcement learning (RL) approaches, *constrained policy optimization* methods (Altman, 2021; Achiam et al., 2017) and *safety-aware reward shaping* methods (Westenbroek et al., 2022; Cao et al.) typically formulate safety requirements as *soft constraints*, allowing temporary violations during learning in exchange for improved long-term performance. An advantage of this formulation is that it preserves smooth gradient propagation during policy optimization. While effective in some settings, this paradigm implicitly assumes that the agent learns to be safe through experience with safety violations, which may be unacceptable in safety-critical applications.

In contrast, *safety filter-based* methods (Hsu et al., 2024; Zhong et al., 2023; Alshiekh et al., 2018; Cheng et al., 2019b) enforce safety through explicit intervention mechanisms that preemptively modify unsafe actions. While such hard enforcement can provide stronger formal safety guarantees, it may introduce abrupt action corrections (Hsu et al., 2024) that disrupt gradient flow and destabilize online and continual learning, particularly under frequent interventions. Recent work has explored differentiable safety filters (Amos & Kolter, 2017; Xiao et al., 2023; Jin et al., 2021; Markgraf et al., 2025; Suttle et al., 2024) that have helped address the gradient-flow issue in safe learning. However, many existing approaches still rely on expensive online optimization or are mainly evaluated in simplified or offline settings (Xiao et al., 2023; Suttle et al., 2024). As a result, scalability to higher-dimensional systems remains an important challenge (Hsu et al., 2024), particularly in safe online learning scenarios that require efficient safety intervention in time.

To resolve the fundamental trade-off between smooth policy optimization and strict safety enforcement, we propose *AutoSafe*, a simple yet effective *safety-aware policy architecture* that combines strong real-time safety assurance with smooth learning dynamics.

*AutoSafe* introduces two key innovations compared to existing safe RL approaches. First, it incorporates risk monitoring and safe intervention as *structural inductive biases* through a differentiable policy composition. Rather than treating safety as an external post-processing step (Hsu et al., 2024), AutoSafe integrates safety awareness directly into the action-generation process, thereby reshaping the policy parameterization and its learning dynamics. Crucially, the differentiable composition allows learning signals to backpropagate through the safety mechanism, enabling stable and efficient policy optimization under safety constraints. As a result, the resulting policy exhibits inherently safety-aware interactions with the environment: it behaves conservatively in high-risk regions while gradually releasing safety constraints and reverting to nominal performance-driven behavior as risk diminishes.

Second, in contrast to typical *last-moment* intervention mechanisms used in safety filters, where corrective actions are applied only when a safety condition is triggered, inducing hard intervention. AutoSafe embeds a *safe policy prior* defined over the entire state space and explicitly guides the intervention process. This prior provides a reliable fallback behavior for safety enforcement and can be constructed using well-established safe controller design methods (Freeman & Kokotovic, 2008; Grandia et al., 2021; Sha, 2001). By supplying well-defined safe actions at all times, AutoSafe enables *precautionary* interventions before the system reaches the safety boundary, resulting in smoother state trajectories and more stable online learning.

We compare AutoSafe against representative safety filter–based approaches and safe reinforcement learning baselines across multiple simulated benchmarks. AutoSafe consistently exhibits stable learning dynamics, achieves strong safety assurance as standard safety filters, and matches or outperforms state-of-the-art safe learning methods in task performance. We further show the practical value of AutoSafe through a real-world cart-pole training experiment.

Our contributions are summarized as follows: (i) We propose a safety-aware policy architecture that embeds safety monitoring and intervention as *structural inductive biases*, enabling smooth action generation and stable online learning. (ii) We provide theoretical insights into key properties of the proposed architecture, including its smooth intervention behavior and learning dynamics. (iii) We empirically evaluate AutoSafe against representative safety filters and safety-aware learning baselines in online policy learning settings across a suite of simulated continuous-control tasks. (iv) We demonstrate the practical applicability of AutoSafe through real-world experiments on a cart-pole system.

## 2 Preliminaries

### 2.1 Policy Learning under Hard Safety Constraints

We formulate the safe learning problem as an infinite-horizon discounted Markov decision process (MDP), defined as $\mathcal{M} = \{\mathcal{S}, \mathcal{A}, P, R, \gamma\}$. Here, $\mathcal{S} \subseteq \mathbb{R}^n$ denotes the $n$-dimensional state space, $\mathcal{A} \subseteq \mathbb{R}^m$ denotes the $m$-dimensional action space, $P : \mathcal{S} \times \mathcal{A} \to \mathcal{S}$ is the state transition function, $R : \mathcal{S} \times \mathcal{A} \to \mathbb{R}$ is a bounded reward function, and $\gamma \in (0, 1)$ is the discount factor.

The objective of safe deep reinforcement learning under hard safety constraints is to find a policy $\pi^\theta : \mathcal{S} \to \mathcal{A}$ that maximizes the expected discounted return while ensuring constraint satisfaction at all time steps:

$$\max_{\pi^\theta} \quad V^{\pi^\theta}(\mathbf{s}) = \mathbb{E}_{\tau \sim \pi^\theta} \left[ \sum_{t=0}^{\infty} \gamma^t R(\mathbf{s}_t, \mathbf{a}_t) \,\middle|\, \mathbf{s}_0 = \mathbf{s} \right], \tag{1}$$

$$\text{s.t.} \quad \mathbf{s}_t \in \mathcal{S}_c, \quad \forall t \in \mathbb{N}. \tag{2}$$

Here, $\tau = \{\mathbf{s}_0, \mathbf{a}_0, \mathbf{s}_1, \mathbf{a}_1, \ldots\}$ denotes the trajectory induced by policy $\pi^\theta$, and $\mathcal{S}_c \subseteq \mathcal{S}$ denotes the set of admissible (safe) states. Throughout this work, we consider a fully observable MDP with continuous state and action spaces.

### 2.2 Safety Filter

A safety filter is an automatic mechanism that continuously monitors the state of an autonomous system and intervenes by modifying intended actions when safety risks are detected (Hsu et al., 2024). The goal of

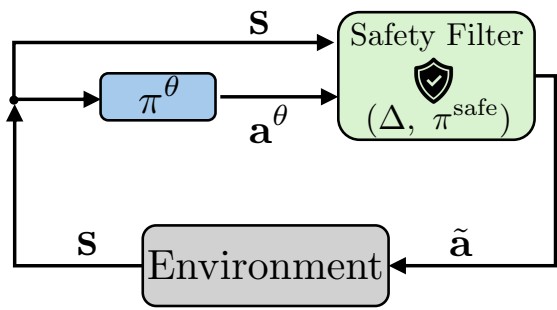

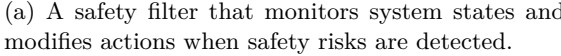

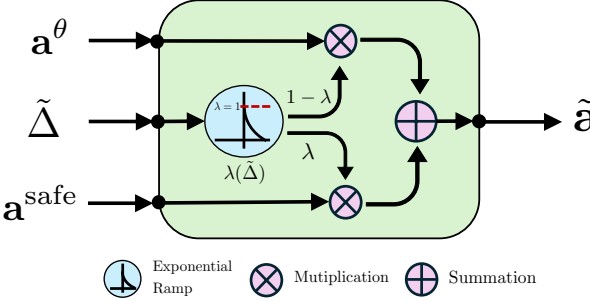

(a) A safety filter that monitors system states and modifies actions when safety risks are detected.

(b) AutoSafe policy architecture with differentiable convex composition of a learning-based policy and a certified safe policy via a state-dependent risk-aware weight $\lambda$.

Figure 1: Diagram of a conventional safety filter architecture and the proposed *AutoSafe* policy architecture.

a safety filter is to enforce hard safety constraints during interaction, while allowing a learning-based policy to optimize task performance whenever possible. As shown in Fig. 1a, typically, a safety filter consists of three components. First, a *safety monitor* $\Delta : \mathcal{S} \times (\mathcal{A}) \to \mathbb{R}$ evaluates the risk associated with a given state and (possibly) intended action by producing a scalar safety margin measurement. Second, a *fallback (safe) policy* $\pi^{\text{safe}} : \mathcal{S} \times (\mathcal{A}) \to \mathcal{A}$ provides safe actions based on the state and the (possibly) intended unsafe action, which is usually deterministic and designed to keep the system state within the constraint set $\mathcal{S}_c$. Finally, an *intervention mechanism* $\eta : \mathcal{S} \times \mathcal{A} \to \mathcal{A}$ determines when and how the fallback policy is applied.

Most existing work on safety filters focuses on the design of safe policies $\pi^{\text{safe}}(\cdot)$, including *action replacement* (Sha, 2001; Zhong et al., 2023; Phan et al., 2020; Alshiekh et al., 2018) and *action projection* (Cheng et al., 2019a; Ames et al., 2019; Grandia et al., 2021), as well as on the design of the safety monitor $\Delta(\cdot)$, which typically falls into *value-based* (Ames et al., 2019; Phan et al., 2020) or *rollout-based* (Chen et al., 2021; Bastani, 2021) approaches.

In contrast, relatively little attention has been paid to the design of the intervention mechanism itself. As a result, most existing safety filters adopt a binary intervention logic, either explicitly via action replacement or implicitly via optimization-based action projection, of the following form:

$$\eta(\mathbf{s}_t, \mathbf{a}_t^\theta) = \begin{cases} \mathbf{a}_t^\theta \sim \pi^\theta(\cdot \mid \mathbf{s}_t), & \text{if } \Delta(\mathbf{s}_t, \mathbf{a}_t^\theta) > \Delta_{\min}, \\ \mathbf{a}_t^{\text{safe}} = \pi^{\text{safe}}(\mathbf{s}_t), & \text{otherwise.} \end{cases}$$

Here, $\mathbf{a}_t^\theta$ denotes the action generated by the learning-based policy $\pi^\theta$. $\Delta(\mathbf{s}_t, \mathbf{a}_t)$ denotes the safety margin at time step $t$, where the safety condition $\Delta(\mathbf{s}_t, \mathbf{a}_t^\theta) > \Delta_{\min}$ is induced by a state–action–dependent safety certificate, such as a control barrier function, or by a state-dependent safety measure $\Delta(\mathbf{s}_t)$ defined with respect to the closed-loop dynamics, e.g., a Lyapunov-based safety value. Satisfying this condition ensures that the system remains within a predefined safe set.

## 3 Safety-filtered Actor-Critic DRL

A key motivation for AutoSafe follows from the fact that online DRL interacts with the environment through *samples* from a stochastic policy. With a conventional safety filter, unsafe samples are handled by a hard intervention to a deterministic safety fallback, which indeed enforces safety but creates an undesirable learning behavior: whenever the filter triggers, the executed action becomes *independent* of $\pi^\theta$. As a result, the agent receives little learning signal about *how* to update the policy to avoid future safety intervention.

To be more specific, most DRL algorithms for continuous control adopt an actor–critic architecture, in which a parameterized policy (actor) is optimized using learning signals from one or more critic networks. Let's denote the policy as $\pi^\theta : \mathcal{S} \to \mathcal{A}$ and a function of some form of critic parameterized by $\psi$ for $\pi^\theta$ as $C_{\pi^\theta}^\psi : \mathcal{S} \times \mathcal{A} \to \mathbb{R}$. The update gradient of the actor in a normal case without safety filter can be generally

expressed as:

$$\nabla_\theta J(\theta) = \mathbb{E}_{\mathbf{s} \sim \mathcal{D}^{\pi^\theta}} \left[ \nabla_\theta \pi^\theta(\mathbf{s}) \, \nabla_{\mathbf{a}^\theta} C_{\pi_\theta}^\psi(\mathbf{s}, \mathbf{a}^\theta) \right], \tag{3}$$

where $\mathcal{D}^{\pi^\theta}$ denote samples generated by $\pi^\theta$ under the dynamic transition $\mathbf{s}' \sim P(\cdot \mid \mathbf{s}, \mathbf{a}^\theta)$. The specific form of the critic $C_{\pi_\theta}^\psi$ depends on the underlying actor-critic algorithm, such as advantage estimation with a state-value critic (Schulman et al., 2017), action-value critics (Lillicrap et al., 2015; Fujimoto et al., 2018), or entropy-regularized action-value critics (Haarnoja et al., 2018).

When a safety filter is applied to the interaction process, the action sampled from the policy $\mathbf{a}^\theta \sim \pi^\theta(\mathbf{s})$ is modified according to a safety intervention rule $\eta(\cdot)$, yielding the executed action $\tilde{\mathbf{a}} = \eta(\mathbf{s}, \pi^\theta(\mathbf{s}))$. We denote the resulting safety-filtered policy by $\tilde{\pi}^\theta$. Under this intervention, the actor update in Eq. 3 becomes

$$\begin{aligned} \nabla_\theta J(\theta) &= \mathbb{E}_{\mathbf{s} \sim \mathcal{D}^{\tilde{\pi}^\theta}} \left[ \nabla_\theta \tilde{\pi}^\theta(\mathbf{s}) \, \nabla_{\tilde{\mathbf{a}}} C_{\tilde{\pi}_\theta}^\psi(\mathbf{s}, \tilde{\mathbf{a}}) \right], \\ &= \mathbb{E}_{\mathbf{s} \sim \mathcal{D}^{\tilde{\pi}^\theta}} \left[ \nabla_\theta \eta(\mathbf{s}, \pi^\theta(\mathbf{s})) \, \nabla_{\tilde{\mathbf{a}}} C_{\tilde{\pi}_\theta}^\psi(\mathbf{s}, \tilde{\mathbf{a}}) \right]. \end{aligned} \tag{4}$$

From Eq. 4, it is evident that when the safety intervention rule $\eta$ is non-differentiable, the safety filter blocks the gradient backpropagation to the actor, preventing effective policy updates.

One possible workaround is to treat the safety filter as part of the environment dynamics (Markgraf et al., 2025), such that the actor update formally retains the same structure as in Eq. 3. This modeling choice, however, induces a modified transition kernel when safety intervention occurs: $P^\eta = P(\mathbf{s}' \mid \mathbf{s}, \eta(\mathbf{s}, \pi^\theta(\mathbf{s})))$, where the next state is not *uniquely* dependent on the action output of $\pi^\theta$.

When safety interventions occur, distinct intended unsafe actions may map to the same executed filtered action through $\eta(\cdot)$. This many-to-one mapping yields identical transition outcomes for different $(\mathbf{s}, \mathbf{a}^\theta)$ pairs, introducing ambiguity in the critic's learning and weakening its ability to capture action-dependent value differences (Markgraf et al., 2025). Consequently, value estimates may become biased and policy gradients less informative, slowing convergence.

Moreover, hard safety interventions often rely on last-moment switching between the learned policy and a fallback controller. The abrupt change may introduce non-smooth transition dynamics near the safety decision boundary. These discontinuities further degrade learning by inflating gradient variance and biasing value estimation in actor–critic methods (Achiam et al., 2017; Ray et al., 2019).

These considerations motivate the design of a safe online learner that (i) transitions smoothly between regimes with inactive and active safety intervention, thereby preserving informative gradients and stable learning signals for policy optimization, and (ii) learns a natural safe behavior by acting *cautiously* as the safety margin tightens: the policy should progressively bias its actions toward safe behavior while reducing exploratory variance to mitigate accidental safety violations.

## 4 Smooth Intervention via Convex Policy Composition

We introduce *AutoSafe* (Fig. 1b), a safety-aware policy architecture that integrates safety directly into action generation. Rather than treating safety as a rejection or correction step applied after action selection, AutoSafe structures the executable policy as a differentiable convex composition of a performance-driven learner and a safety prior.

### 4.1 Structural Policy Composition

At each time step, the executed action $\tilde{\mathbf{a}}$ is synthesized by interpolating between the stochastic action proposed by the learning policy, $\mathbf{a}^\theta \sim \pi^\theta(\cdot \mid \mathbf{s})$, and a deterministic safe action provided by the safety prior, $\mathbf{a}^{\text{safe}} = \pi^{\text{safe}}(\mathbf{s})$:

$$\tilde{\mathbf{a}} = (1 - \lambda(\mathbf{s})) \, \mathbf{a}^\theta + \lambda(\mathbf{s}) \, \mathbf{a}^{\text{safe}}, \quad \lambda(\mathbf{s}) \in [0, 1]. \tag{5}$$

Here, $\lambda(s)$ acts as a dynamic mixing that balances between reward maximization and risk mitigation.

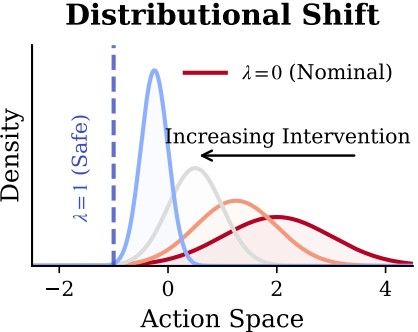
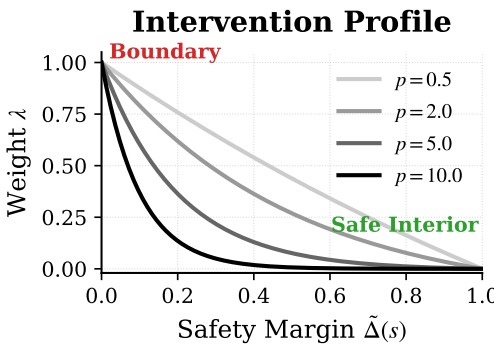

Figure 2: **(Left)** Distributional Shaping (Sec. 4.2): As the intervention weight $\lambda$ increases, the policy distribution gradually shifts its mean toward the safe anchor $a^{\text{safe}}$, while its variance decreases quadratically ( Eq. 6). **(Right)** Learnable Sharpness (Sec. 4.4): The sharpness parameter $p$ controls the intervention profile. A larger $p$ creates a "sharper" boundary that delays intervention to prioritize performance, while a smaller $p$ induces earlier and more gradual intervention.

This design leverages the geometry of continuous control tasks. Assuming the feasible action space $\mathcal{A}$ is convex (e.g., standard box constraints), any convex combination of two valid actions is guaranteed to remain valid ($\mathbf{a}^\theta, \mathbf{a}^{\text{safe}} \in \mathcal{A} \to \tilde{\mathbf{a}} \in \mathcal{A}$ (Boyd & Vandenberghe, 2004)). Physically, this means the intervention does not arbitrarily "clip" the action, but rather pulls it along the line segment connecting the agent's desired action to the safe anchor.

This formulation secures two key properties for online learning: **Differentiability:** Unlike discrete safety filters, Eq. 5 is fully differentiable with respect to $\lambda$. This preserves gradient flow even during intervention. **Expressiveness:** The composite policy strictly contains the original policy class. In safe regions ($\lambda \to 0$), the agent recovers full autonomy to learn optimal behaviors; as risk escalates ($\lambda \to 1$), behavior smoothly collapses to the safe prior.

## 4.2 Distributional Shaping & Variance Control

The structural composition in Eq. 5 fundamentally reshapes the policy's stochastic profile. By viewing the intervention as a transformation (a pushforward under the affine map) of the learner's distribution $\pi^\theta(\cdot \mid \mathbf{s})$, we can quantify exactly how the geometric interpolation alters the agent's behavior.

For Gaussian policies, standard in continuous control where $\pi^\theta(\cdot \mid \mathbf{s}) = \mathcal{N}(\mu_\theta(\mathbf{s}), \Sigma_\theta(\mathbf{s}))$, the executed policy $\tilde{\pi}$ remains Gaussian with moments:

$$\tilde{\mu}(\mathbf{s}) = (1 - \lambda)\mu_\theta(\mathbf{s}) + \lambda \mathbf{a}^{\text{safe}}, \quad \tilde{\Sigma}(\mathbf{s}) = (1 - \lambda)^2 \Sigma_\theta(\mathbf{s}). \tag{6}$$

This result reveals that the convex structure of *AutoSafe* automatically couples safety with uncertainty reduction. As $\lambda$ increases, the intervention induces:

- **Mean bias:** The expected action $\tilde{\mu}$ shifts toward the safe prior $\mathbf{a}^{\text{safe}}$, correcting the learner's intent.

- **Variance damping:** Exploration noise is reduced by a quadratic factor $(1 - \lambda)^2$.

This coupling is the first-principles mechanism that stabilizes learning: as the agent approaches risk (high $\lambda$), it does not merely steer away; it effectively restricts stochastic exploration to prevent accidental boundary violations.

## 4.3 Risk-Calibrated Learnable Intervention

With the intervention mechanism in Sec. 4.1 and its variance-damping effect in Sec. 4.2 established, we now specify how the intervention weight $\lambda(\mathbf{s})$ is chosen. Our guiding principle is *minimal intervention*: the

learning policy should retain as much control authority as possible, while the executed policy should place only negligible probability mass in unsafe regions. For the smooth rate analyzed below, $\lambda(\mathbf{s})$ is treated as fixed after conditioning on $\mathbf{s}$, which preserves the Gaussian pushforward identity in Eq. 6. Action-dependent safety checks are still allowed, but are used only as hard triggers.

Let

$$\hat{\Delta}(\mathbf{s}, \mathbf{a}) := \Delta(\mathbf{s}, \mathbf{a}) - \Delta_{\min} \tag{7}$$

denote the relative safety margin, so that $\hat{\Delta}(\mathbf{s}, \mathbf{a}) \geq 0$ indicates satisfaction of the safety condition. For a fixed state $\mathbf{s}$, the executed action $\tilde{\mathbf{a}}_\lambda$ induces the random margin

$$Z(\lambda) := \hat{\Delta}(\mathbf{s}, \tilde{\mathbf{a}}_\lambda).$$

We require the executed policy to satisfy the state-wise chance constraint

$$\Pr(Z(\lambda) < 0 \mid \mathbf{s}) = \Pr(\Delta(\mathbf{s}, \tilde{\mathbf{a}}_\lambda) < \Delta_{\min} \mid \mathbf{s}) \leq \delta(\mathbf{s}), \tag{8}$$

where $\delta(\mathbf{s})$ is a risk tolerance. This constraint bounds the unsafe tail probability of the executed action distribution.

A tractable intervention template follows by locally approximating the relative safety margin as affine along the interpolation segment between the nominal action $\mathbf{a}^\theta$ and the safe prior action $\mathbf{a}^{\mathrm{safe}}$. Under this approximation, and using a one-sided tail condition, the smallest feasible intervention weight has the closed form

$$\lambda^*(\mathbf{s}) = \mathrm{clip}_{[0,1]} \left( \frac{\beta_\delta \sigma_\Delta(\mathbf{s}) - \mu_\Delta(\mathbf{s})}{\Delta_{\mathrm{safe}}(\mathbf{s}) - \mu_\Delta(\mathbf{s}) + \beta_\delta \sigma_\Delta(\mathbf{s})} \right), \tag{9}$$

where

$$\mu_\Delta(\mathbf{s}) = \mathbb{E}\left[ \hat{\Delta}(\mathbf{s}, \mathbf{a}^\theta) \mid \mathbf{s} \right], \qquad \sigma_\Delta(\mathbf{s}) = \mathrm{Std}\left[ \hat{\Delta}(\mathbf{s}, \mathbf{a}^\theta) \mid \mathbf{s} \right],$$

and

$$\Delta_{\mathrm{safe}}(\mathbf{s}) = \hat{\Delta}\left( \mathbf{s}, \mathbf{a}^{\mathrm{safe}}(\mathbf{s}) \right).$$

Here $\beta_\delta$ is a risk quantile, e.g., $\beta_\delta = \Phi^{-1}(1 - \delta(\mathbf{s}))$ for Gaussian margins, or the conservative sub-Gaussian choice $\beta_\delta = \sqrt{2 \log(1/\delta(\mathbf{s}))}$. The derivation and approximation assumptions are given in Appendix A.

Eq. 9 provides the key design insight. If the risk-adjusted nominal margin

$$L_\theta(\mathbf{s}) := \mu_\Delta(\mathbf{s}) - \beta_\delta \sigma_\Delta(\mathbf{s})$$

is nonnegative, the nominal policy already satisfies the tail-risk condition and no intervention is needed. Otherwise, the numerator in Eq. 9 measures the nominal safety deficit, while the denominator measures the safety improvement available by moving toward the safe prior. Thus, $\lambda$ should increase as the safety margin decreases, as uncertainty increases, or as the desired risk tolerance becomes stricter.

In online deep RL, directly estimating $\sigma_\Delta(\mathbf{s})$ and selecting a state-dependent $\beta_\delta$ can be unreliable. AutoSafe therefore uses Eq. 9 as a structural template rather than as an online estimator. The implemented rate enforces three properties: monotonicity with respect to the safety margin, endpoint safety at the boundary, and a learnable sharpness controlling how quickly intervention rises.

The monitor provides a normalized state-conditioned margin $\tilde{\Delta}(\mathbf{s}) \in [0,1]$, where $\tilde{\Delta}(\mathbf{s}) = 1$ denotes a clearly safe interior state and $\tilde{\Delta}(\mathbf{s}) = 0$ denotes the safety boundary. We instantiate the intervention weight as

$$\lambda\big(\tilde{\Delta}(\mathbf{s}), p(\mathbf{s})\big) = \frac{\exp\big(p(\mathbf{s})(1 - \tilde{\Delta}(\mathbf{s}))\big) - 1}{\exp(p(\mathbf{s})) - 1}, \tag{10}$$

where $p(\mathbf{s}) > 0$ is a learnable sharpness parameter. This map satisfies

$$\lambda(1, p) = 0, \qquad \lambda(0, p) = 1, \qquad \frac{\partial \lambda}{\partial \tilde{\Delta}} < 0.$$

Hence, intervention vanishes in the safe interior and increases monotonically as the system approaches the safety boundary.

The sharpness parameter controls the conservatism of this transition. Smaller $p(\mathbf{s})$ produces earlier and smoother intervention, while larger $p(\mathbf{s})$ delays intervention until closer to the boundary, preserving more autonomy for the learning policy. We parameterize it using a small prediction head,

$$p(\mathbf{s}; \phi) = p_{\min} + (p_{\max} - p_{\min}) \operatorname{sigmoid}(h_\phi(\mathbf{s})), \qquad 0 < p_{\min} < p_{\max}. \tag{11}$$

The parameters $\phi$ are optimized jointly with the actor through the executed action

$$\tilde{\mathbf{a}} = (1 - \lambda(\mathbf{s}))\mathbf{a}^\theta + \lambda(\mathbf{s})\mathbf{a}^{\text{safe}}.$$

Finally, near-boundary determinism follows directly from Eq. 6:

$$\tilde{\Sigma}(\mathbf{s}) = (1 - \lambda(\mathbf{s}))^2 \Sigma_\theta(\mathbf{s}).$$

As $\lambda(\mathbf{s}) \to 1$, the executed covariance collapses, suppressing exploratory variance near the safety boundary. If the monitor detects an imminent safety violation, e.g., $\hat{\Delta}(\mathbf{s}, \mathbf{a}^\theta) \leq 0$ for a state-action monitor or $\hat{\Delta}(\mathbf{s}) \leq 0$ for a state monitor, then $\tilde{\Delta}(\mathbf{s}) = 0$, and *AutoSafe* sets $\lambda = 1$ to fully execute the certified safe action $\mathbf{a}^{\text{safe}}$. Thus, the learnable rate shapes precautionary behavior before the boundary is reached, while the hard trigger preserves the endpoint safety inherited from the certified safe prior.

## 5 Experiments and Results

### 5.1 Backbone Algorithms

The proposed policy composition diagram is *safety-filter agnostic*. As a representative instantiation, we adopt the widely used *Simplex* architecture (Sha, 2001) as the baseline safety design. Simplex employs a robust, *certified-safe fallback policy* obtained by solving an offline model-based optimization problem that explicitly accounts for closed-loop stability under safety constraints.

The safe action is given by a state-feedback control law $\mathbf{a}^{\text{safe}} \coloneqq \mathbf{Fs}$, where $\mathbf{F}$ denotes the feedback gain matrix. Safety monitoring is performed using a state-dependent Lyapunov-like function $\Delta(\mathbf{s}) \coloneqq 1 - \mathbf{s}^\top \mathbf{Ps}$, where $\mathbf{P} \succ 0$ is a positive definite Lyapunov matrix associated with the closed-loop dynamics under the fallback controller, obtained via standard Lyapunov or LMI-based synthesis. Following the convention introduced in Section 2.2, control switches to $\pi^\theta$ when $\Delta(\mathbf{s}) > \Delta_{\min}$ and to $\pi^{\text{safe}}$ otherwise.

As illustrated in a two-dimensional setting in Fig. 3, the condition $\Delta(\mathbf{s}) > \Delta_{\min}$ defines an inner safe region.

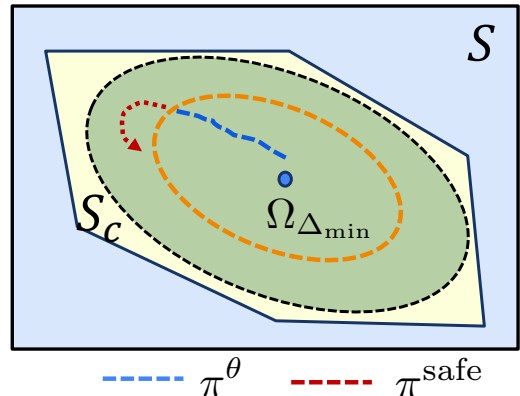

Figure 3: Illustration of a Simplex-style safety mechanism with an inner safe set $\Omega_{\Delta_{\min}}$ and switching to a certified safe policy at the safety boundary in a static 2D case.

$$\Omega_{\Delta_{\min}} \coloneqq \left\{ \mathbf{s} \mid 1 - \mathbf{s}^\top \mathbf{Ps} > \Delta_{\min} \right\},$$

shown as the ellipsoidal set with yellow boundary. When the system state lies within $\Omega_{\Delta_{\min}}$, control authority is assigned to the learning-based policy $\pi^\theta$. Once the monitored safety margin $\Delta(\mathbf{s})$ decreases to the threshold $\Delta_{\min}$, the controller immediately switches to the certified safe policy $\pi^{\text{safe}}$. The black dashed boundary denotes the largest invariant safety envelope, corresponding to the zero level set $\Delta(\mathbf{s}) = 0$ (i.e., $\mathbf{s}^\top \mathbf{Ps} = 1$), within which the fallback controller guarantees forward invariance for safety. Notably, Simplex avoids solving optimization problems at runtime as in (Ames et al., 2019; Cheng et al., 2019b) and always provides a valid safe action as a backup, making it particularly attractive for real-time safety-critical applications.

In this work, we adopt *soft-actor-critic* algorithm (SAC) (Haarnoja et al., 2018) for policy learning, where the policy parameters $\theta$ are optimized via gradient ascent on the objective

$$J(\theta) = \mathbb{E}_{\mathbf{s} \sim \mathcal{D}, \tilde{\mathbf{a}} \sim \tilde{\pi}^\theta} \left[ Q^\psi(\mathbf{s}, \tilde{\mathbf{a}}) - \alpha \log \tilde{\pi}^\theta(\tilde{\mathbf{a}} \mid \mathbf{s}) \right], \tag{12}$$

where $Q^\psi$ denotes the action-value critic that estimates the expected return of executing the composite policy $\tilde{\pi}^\theta$, i.e., taking action $\tilde{\mathbf{a}} \sim \tilde{\pi}^\theta$ as defined in Eq. 5, and $\alpha$ is the temperature parameter that weights the entropy regularization term. AutoSafe does not modify the critic or the learning objective. Instead, it alters only how the actor produces executable actions by composing the learning-based action with a safe fallback action. The remainder of the interaction and training scheme remains identical to the original SAC algorithm.

## 5.2 Experiment Setup

We evaluate our method against safety-filter-based baselines and safe RL baselines in the online safe learning context, including: **Safety Filter:** i) SimplexRL (Alshiekh et al., 2018; Phan et al., 2020; Cai et al., 2025): runtime safety filter by safe action replacement based on *Simplex*; ii) CBF (Ames et al., 2019; Cheng et al., 2019a): runtime safety filter by safe action projection using control barrier function. **Safe Learning with Prior:** i) AdaLam (Cheng et al., 2019c; Tian et al., 2024): weighted summation of safe policy prior and DRL policy, where the weights are adaptively adjusted based on context and safety; ii) Residual (Johannink et al., 2019): learns a residual policy on top of a fixed safe policy prior; **Constrained RL:** i) Lyapunov (Westenbroek et al., 2022; Cao et al.): adds Lyapunov-based penalties for reward shaping for safe exploration; ii) Lagrangian (Ha et al., 2020; Achiam et al., 2017): constrained policy optimization via dual variables.

We consider four simulated and one real-world case studies, including: i) CartPole Balancing (sim & real) (Towers et al., 2024), with continuous force control under position and angle constraints; ii) Glucose Regulation (Tian et al., 2024), with continuous insulin control under glucose level constraints; iii) 3D Quadrotor Goal Reaching (Yuan et al., 2022), with four-dimensional thrust control under position and attitude constraints; and iv) Quadruped Navigation (Yang et al., 2022a), with six-dimensional acceleration control on uneven terrain under height and velocity constraints. Implementation details are summarized in Appendix C. All experiments are run with five random seeds.

## 5.3 Main Results

Table 1: **Main Results.** Performance and safety comparison across four continuous control tasks. We report mean $\pm$ std over five random seeds. Higher return is better ($\uparrow$), while fewer safety violations are better ($\downarrow$). Best results are marked in **bold**.

| | Cartpole | | Glucose | | 3D Quadrotor | | Quadruped | |
|---|---|---|---|---|---|---|---|---|
| Method | Return ↑ | Viol. ↓ | Return ↑ | Viol. ↓ | Return ↑ | Viol. ↓ | Return ↑ | Viol. ↓ |
| SAC (Haarnoja et al., 2018) | **478.6 ± 2.4** | 74.4 ± 4.4 | -155.0 ± 12.4 | 24.4 ± 17.2 | 0.6 ± 0.9 | 2437.2 ± 625.3 | 601.5 ± 330.5 | 1895.6 ± 257.1 |
| SimplexRL (Phan et al., 2020) | 384.5 ± 127.9 | **0.0 ± 0.0** | **-124.0 ± 4.1** | **0.0 ± 0.0** | 17.5 ± 10.7 | 50.0 ± 53.7 | 1880.5 ± 461.8 | 0.2 ± 0.4 |
| CBF (Ames et al., 2019) | 448.2 ± 5.0 | 43.8 ± 10.3 | -143.4 ± 10.8 | 13.4 ± 13.6 | 132.4 ± 123.4 | 876.6 ± 161.7 | 12.9 ± 2.9 | 17502.0 ± 2604.0 |
| AdaLam (Tian et al., 2024) | 473.7 ± 2.4 | 39.8 ± 89.0 | -384.1 ± 24.1 | 0.0 ± 0.0 | 88.8 ± 198.1 | 47149.6 ± 49797.1 | 477.4 ± 296.2 | 960.6 ± 224.2 |
| Residual (Johannink et al., 2019) | 477.6 ± 1.7 | 16.4 ± 11.2 | -503.3 ± 3.6 | 47.8 ± 85.1 | 0.4 ± 0.8 | 1258.8 ± 238.2 | 1493.5 ± 338.5 | 522.6 ± 57.6 |
| Lyapunov (Westenbroek et al., 2022) | 475.2 ± 2.9 | 19.0 ± 10.4 | -195.6 ± 33.1 | 20.2 ± 14.8 | 0.1 ± 0.1 | 8391.0 ± 14100.4 | 408.5 ± 178.7 | 2082.4 ± 989.3 |
| Lagrangian (Ha et al., 2020) | 474.9 ± 4.4 | 99.2 ± 27.4 | -192.6 ± 12.5 | 23.6 ± 16.4 | 0.0 ± 0.1 | 262756.2 ± 62636.0 | 17.9 ± 8.4 | 17824.6 ± 3828.9 |
| **AutoSafe (Ours)** | 477.7 ± 2.5 | **0.0 ± 0.0** | -131.2 ± 3.7 | **0.0 ± 0.0** | **740.8 ± 27.2** | **0.0 ± 0.0** | **2015.2 ± 757.7** | **0.0 ± 0.0** |

Overall, AutoSafe outperforms the baselines in the majority of tasks and runs, achieving higher returns while maintaining strong safety assurance, as shown in Table 1. In relatively low-dimensional settings such as Cartpole and Glucose, hard-intervention–based safety filters (e.g., SimplexRL and CBF) achieve performance similar to AutoSafe. However, their performance degrades substantially on higher-dimensional, more dynamically complex tasks, such as Quadrotor and Quadruped. In these environments, frequent safety interventions introduce abrupt distribution shifts that hinder smooth function approximation, leading to unstable learning and occasional divergence.

Residual learning combines learning-based and safe actions through direct summation and performs well in simpler tasks such as Cartpole and Glucose. However, its effectiveness depends strongly on the quality and

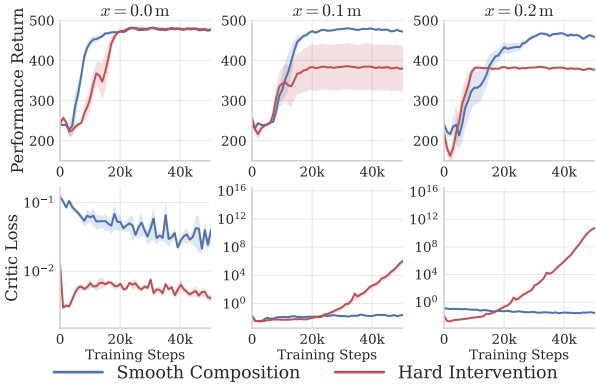

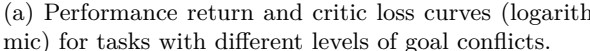

(a) Performance return and critic loss curves (logarithmic) for tasks with different levels of goal conflicts.

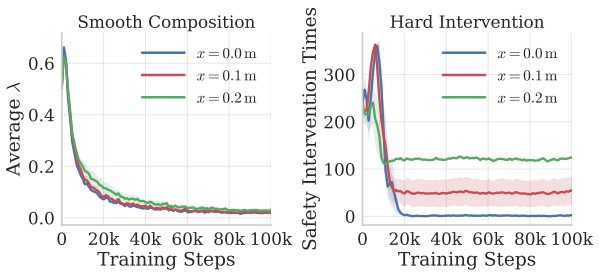

(b) Learning to Be Safe: Smooth vs. Hard Intervention. **(Left)**: Smooth policy composition yields stable and consistent safety learning across different levels of task conflict. **(Right)**: Hard safety intervention initially reduces interventions but later stalls, leading to repeated safety interventions.

Figure 4: Comparison of learning dynamics and safety intervention behavior.

alignment of the safe prior; mismatches can bias learning and slow convergence in more complex domains. Similarly, AdaLam adaptively blends learning-based and safe actions via a weighting parameter $\lambda$. However, safety is not explicitly enforced, and the weighting parameter $\lambda$ is optimized solely for task performance. In practice, we observe that as $\lambda$ decreases, safety violations increase dramatically, causing learning to make little further progress.

Lastly, approaches that learn using safe priors benefit from incorporating domain knowledge and tend to achieve stronger safety and performance within limited training budgets. In contrast, constrained RL methods typically rely on learning safety from constraint violations, which can lead to increased violations during training and require substantially more interaction to converge. We summarize the computational time comparison for all methods in Table 5 at Section D.7, showing that *AutoSafe* achieves computational efficiency comparable to vanilla SAC.

### 5.4 Influences of hard intervention on learning

In this section, we study the impact of *hard safety interventions* on learning dynamics using a controlled CartPole experiment based on *Simplex*. We consider a position-tracking task in which the learning policy must stabilize the system while reaching a target cart position. The safe policy regulates the system toward $x_{\text{goal}} = 0.0$ m. To induce different levels of goal conflict, the learning policy is tasked with reaching target positions $x_{\text{goal}} \in \{0.0, 0.1, 0.2\}$ m, with larger offsets corresponding to stronger conflicts between task objectives and safety enforcement.

As shown in Fig. 4a, our method robustly handles different levels of goal conflict and consistently outperforms the *hard-intervention* baseline. In contrast, learning under the hard-intervention mechanism becomes increasingly sensitive as the goal conflict intensifies. Frequent safety interventions lead to unstable critic estimation, manifested by rapidly increasing critic loss. As a result, policy optimization stagnates, and overall task performance degrades significantly.

Fig. 4b further illustrates the learning dynamics under different goal-reaching tasks. With smooth policy composition, the agent gradually reduces the intervention weight $\lambda$, relying less on the safe policy and increasingly exploiting the learning-based policy to maximize task performance. By contrast, while the hard-intervention mechanism initially reduces the intervention frequency, this trend breaks down under stronger goal conflicts. For $x_{\text{goal}} = 0.1$ m and $0.2$ m, the intervention rate plateaus after approximately 20k steps, coinciding with a sharp rise in critic loss (Fig. 4a). This behavior suggests that excessive and persistent interventions induce extrapolation errors in the critic, preventing the policy from receiving informative learning signals and ultimately halting learning progress.

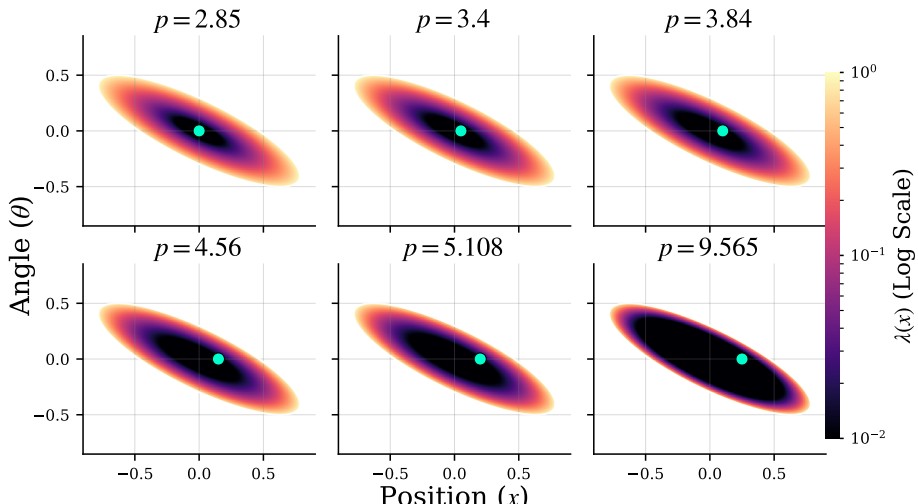

Figure 5: **CartPole safety envelope visualization.** Visualization of $\lambda(\tilde{\Delta}(\mathbf{s}))$ over the $x$ and $\theta$ dimensions for different learned values of $p$ across CartPole tasks. The parameter $p$ adapts to the level of goal conflict: after the agent has learned to remain safe, increasing $p$ reduces $\lambda$ at the same state, downweighting the safety policy to favor higher task performance within the safety envelope.

Further analysis on the sensitivity to the choice of $\Delta_{\min}$ for both SimplexRL and *AutoSafe* is provided in Table 4 at the D.6. The results show the expected trade-off: a larger $\Delta_{\min}$ leads to earlier safety intervention and more conservative behavior, while a smaller $\Delta_{\min}$ allows more aggressive exploration at the cost of increased safety violations.

## 5.5  Adaptation of learned $p$

Making the sharpness parameter $p$ learnable enables the agent to adaptively adjust its value, thereby maximizing performance across different system dynamics and tasks. To validate this hypothesis, we visualize the learned $p$ from the a more fine-grained CartPole experiment in Fig. 5. As the goal mismatch increases, the suboptimality of the safe prior becomes more pronounced. Correspondingly, the value of learned $p$ increases, which in turn reduces the blending weight $\lambda$. This indicates that the agent adaptively learns to rely less on the suboptimal safe policy to seek higher return.

We observe that the converged sharpness parameter $p$ varies substantially across tasks (see Fig. 12 at Appendix D) and is not analytically tractable to determine. Heuristic schedules, therefore, require task-specific tuning and may introduce design bias, as seen in the Quadrotor setting (see Fig. 13 in Appendix D). Learning $p$ along with the policy avoids this issue and yields robust performance across all tasks.

## 5.6  Precautionary safety intervention

The leverage of a safety prior enables AutoSafe to perform *precautionary* safety intervention, rather than relying on last-moment triggering. As visualized in Fig. 6, when the system approaches the safety boundary, the agent learns when and how strongly to intervene (e.g., in CartPole, $\lambda$ increases primarily once the safety margin falls below $\approx 0.3$).

Compared to CartPole, the Quadrotor requires more frequent and stronger interventions because the safety margin is more sensitive to state variations in the higher-dimensional state space ($\mathbf{s} \in \mathbb{R}^{12}$ vs. $\mathbf{s} \in \mathbb{R}^{4}$), where deviations across multiple state components can jointly cause the safety margin to decrease more sharply. In both cases, $\lambda$ remains small most of the time, indicating that the learned policy largely maintains performance-driven behavior while being safe.

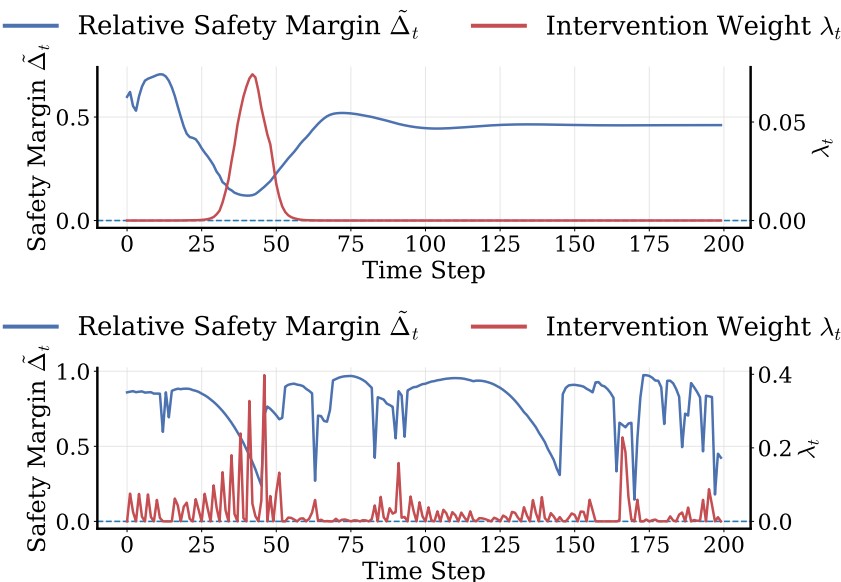

Figure 6: AutoSafe-controlled trajectories for CartPole (**Top**) and Quadrotor (**Bottom**). The intervention weight $\lambda$ rises as the system approaches safety boundaries, enabling precautionary control, and remains low elsewhere to preserve performance-driven behavior.

### 5.7 Real world applicability

The proposed architecture is favorable for safe online policy learning in real-world settings. By smoothly intervening, it avoids abrupt changes in system dynamics that can increase stochasticity and destabilize learning. Moreover, unlike CBF- or MPC-based methods (Cheng et al., 2019a; Grandia et al., 2021), it does not require online optimization, resulting in higher computational efficiency suitable for high-frequency control tasks.

We demonstrate this feature in a real-world CartPole learning task (Fig. 7a), where the AutoSafe policy is deployed on an embedded device (Raspberry Pi) to ensure safe interaction, while policy parameters are periodically updated from a remote workstation. As shown in Fig. 7b, AutoSafe enables the agent to interact safely with the environment while continuously improving task performance. Over time, the parameter $\lambda$ decreases, indicating that the policy gradually reduces its reliance on the safe prior as it learns safer and more performant behaviors. Additionally, we show that AutoSafe can learn safely under dynamically changing obstacle constraints by efficiently adapting the offline-certified safe recoverable region without online optimization, as demonstrated in the additional CartPole and Quadrotor experiments in Section D.5.

## 6 Related Work

**Safety as a soft constraint:** A large body of work improves safety during learning by formulating safety requirements as soft constraints, including constrained policy optimization (Wachi & Sui, 2020; Achiam et al., 2017), policy-prior aided training (Xie et al., 2018), and Lyapunov-based reward shaping (Westenbroek et al., 2022; Zhao et al., 2023; Cao et al.). Safety assurance in these methods is typically asymptotic or expectation-based after convergence, rather than being hard-enforced during training and deployment. An emerging direction incorporates safety priors to guide learning, using residual policies (Johannink et al., 2019), regularization (Cheng et al., 2019c), or policy fusion (Rana et al., 2023). However, these approaches primarily target data efficiency and performance improvement, rather than explicitly enforcing safety constraints.

**Enforce hard safety constraint:** Hard safety guarantees are typically enforced via safety filters (Hsu et al., 2024), including action projection–based methods (Ames et al., 2019; Cheng et al., 2019a) and action

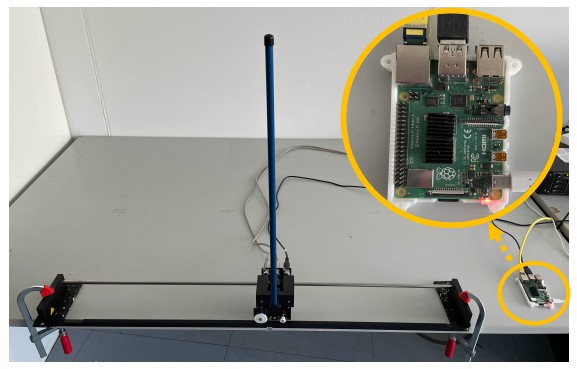

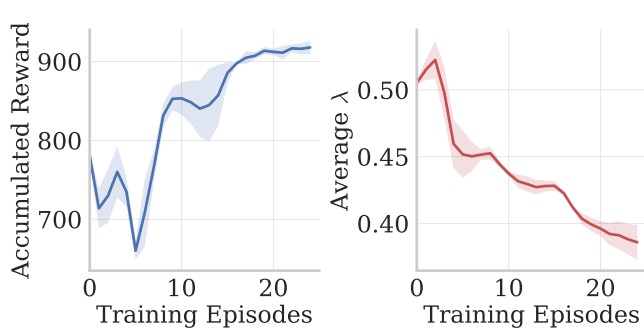

(a) Real-world safe learning on a cart-pole system.

(b) Experimental results over three runs.

Figure 7: Real-world experiments with AutoSafe. The agent is deployed on an embedded device, enabling safe interaction at a control frequency of 50 Hz.

replacement mechanisms (Alshiekh et al., 2018; Zhong et al., 2023; Bak et al.; Nesti et al., 2025). While effective at preventing violations, frequent or abrupt interventions can disrupt gradient flow and destabilize online and continual learning.

Recent efforts on differentiable safety filters (Amos & Kolter, 2017; Xiao et al., 2023; Jin et al., 2021; Markgraf et al., 2025; Suttle et al., 2024) have helped alleviate the gradient-flow issue in safe learning. However, many of these approaches still rely on solving online optimization problems (Jin et al., 2021; Amos & Kolter, 2017), or are primarily evaluated in simplified task Suttle et al. (2024) or offline training settings using pre-collected datasets (Xiao et al., 2023). While these methods demonstrate promising capabilities, their scalability to higher-dimensional systems remains an active challenge (Hsu et al., 2024), which may limit their applicability in safe online learning scenarios where computational efficiency and timely safety intervention are critical.

## 7 Conclusions and Limitations

We introduced *AutoSafe*, a policy architecture that incorporates safety monitoring and intervention directly into the action-generation process to enable smooth online reinforcement learning under hard safety constraints. By integrating safety as a structural inductive bias, the proposed framework preserves recoverable safety while maintaining stable learning dynamics and strong task performance. Experimental results on both simulated and real-world systems demonstrate that *AutoSafe* can provide effective safety assurance without sacrificing learning capability.

Despite these promising results, several limitations remain for more challenging real-world training. First, similar to many provably safe learning architectures Krasowski et al. (2023), *AutoSafe* builds on model-based safety design using prior knowledge of the system dynamics. The quality of the derived safe policy or safety monitor can affect the overall system performance. Although the proposed smooth policy composition is designed to remain robust to sub-optimal safe policies through the adaptive mixing parameter $\lambda$, inaccurate or overly conservative models may still limit exploration capability and achievable task performance.

In addition, the current framework primarily considers safety constraints defined in relatively local and structured state spaces. In more dynamic environments with time-varying constraints, disturbances, or moving obstacles, the safety filter backbone may need to be adapted accordingly, as shown in Section D.5. Maintaining safety in such scenarios may require online adaptation of the safe policy for disturbance rejection (Wang et al., 2013) or safe set (Ames et al., 2019; Wabersich & Zeilinger, 2019).

Future work will therefore focus on extending *AutoSafe* toward more complex real-world settings, including higher-dimensional robotic systems, adaptive safety mechanisms under uncertainty, and dynamic environments with evolving safety constraints.

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

# A  Supplementary Results for Risk-Calibrated Intervention

This appendix provides the assumptions and derivations supporting Sec. 4.3–Sec. 4.3. The purpose is not to require online estimation of all quantities in Eq. 9, but to justify the structure of the intervention rule used by AutoSafe.

## A.1  Setup and Assumptions

Fix a state $\mathbf{s} \in \mathcal{S}$. The learner samples a nominal action $\mathbf{a}^\theta \sim \pi^\theta(\cdot \mid \mathbf{s})$, and the safe prior produces a deterministic action

$$\mathbf{a}^{\mathrm{safe}}(\mathbf{s}) := \pi^{\mathrm{safe}}(\mathbf{s}). \tag{13}$$

For a state-conditioned intervention weight $\lambda \in [0, 1]$, AutoSafe executes

$$\tilde{\mathbf{a}}_\lambda = (1 - \lambda)\mathbf{a}^\theta + \lambda \mathbf{a}^{\mathrm{safe}}(\mathbf{s}). \tag{14}$$

Throughout this appendix, $\lambda$ is treated as fixed after conditioning on $\mathbf{s}$. If $\lambda$ is made dependent on the sampled action $\mathbf{a}^\theta$, then Eq. 14 is no longer an affine map with deterministic coefficient, and the Gaussian pushforward identities below become approximations rather than exact distributional identities.

Assume the nominal policy is Gaussian,

$$\pi^\theta(\cdot \mid \mathbf{s}) = \mathcal{N}(\mu_\theta(\mathbf{s}), \Sigma_\theta(\mathbf{s})). \tag{15}$$

Then $\tilde{\mathbf{a}}_\lambda$ is Gaussian with

$$\tilde{\mathbf{a}}_\lambda \sim \mathcal{N}\big(\tilde{\mu}(\mathbf{s}), \tilde{\Sigma}(\mathbf{s})\big), \qquad \tilde{\mu}(\mathbf{s}) = (1 - \lambda)\mu_\theta(\mathbf{s}) + \lambda \mathbf{a}^{\mathrm{safe}}(\mathbf{s}), \qquad \tilde{\Sigma}(\mathbf{s}) = (1 - \lambda)^2 \Sigma_\theta(\mathbf{s}). \tag{16}$$

This recovers Eq. 6 in the main text. For $0 \le \lambda < 1$, the differential entropy satisfies

$$\mathcal{H}(\tilde{\pi}_\lambda(\cdot \mid \mathbf{s})) = \mathcal{H}\big(\pi^\theta(\cdot \mid \mathbf{s})\big) + m \log(1 - \lambda), \tag{17}$$

where $m$ is the action dimension. At $\lambda = 1$, the executed distribution collapses to a Dirac distribution at $\mathbf{a}^{\mathrm{safe}}(\mathbf{s})$ and does not have finite differential entropy.

Safety is specified by a margin function $\Delta(\mathbf{s}, \mathbf{a})$ and threshold $\Delta_{\min}$. We define the relative margin

$$\hat{\Delta}(\mathbf{s}, \mathbf{a}) := \Delta(\mathbf{s}, \mathbf{a}) - \Delta_{\min}, \qquad \hat{\Delta}(\mathbf{s}, \mathbf{a}) \ge 0 \iff \text{safe at } \mathbf{s}. \tag{18}$$

The random margin under the executed action is

$$Z(\lambda) := \hat{\Delta}(\mathbf{s}, \tilde{\mathbf{a}}_\lambda). \tag{19}$$

The chance constraint used in the main text is

$$\Pr(Z(\lambda) < 0 \mid \mathbf{s}) \le \delta(\mathbf{s}). \tag{20}$$

For convenience, define

$$\mu_\Delta(\mathbf{s}) := \mathbb{E}\Big[\hat{\Delta}(\mathbf{s}, \mathbf{a}^\theta) \mid \mathbf{s}\Big], \qquad \sigma_\Delta^2(\mathbf{s}) := \mathrm{Var}\Big(\hat{\Delta}(\mathbf{s}, \mathbf{a}^\theta) \mid \mathbf{s}\Big), \qquad \Delta_{\mathrm{safe}}(\mathbf{s}) := \hat{\Delta}\big(\mathbf{s}, \mathbf{a}^{\mathrm{safe}}(\mathbf{s})\big). \tag{21}$$

All expectations and variances are conditioned on the fixed state $\mathbf{s}$.

The following assumptions are used only to justify the closed-form template in Eq. 9.

**Assumption A.1.** For fixed $\mathbf{s}$, the relative margin $\hat{\Delta}(\mathbf{s}, \mathbf{a})$ is locally approximated by an affine function of $\mathbf{a}$ on a neighborhood containing the interpolation segment between $\mathbf{a}^\theta$ and $\mathbf{a}^{\mathrm{safe}}(\mathbf{s})$. That is, there exist $b(\mathbf{s}) \in \mathbb{R}$ and $\mathbf{g}(\mathbf{s}) \in \mathbb{R}^m$ such that

$$\hat{\Delta}(\mathbf{s}, \mathbf{a}) = b(\mathbf{s}) + \mathbf{g}(\mathbf{s})^\top \mathbf{a} + \varepsilon_{\mathrm{lin}}(\mathbf{s}, \mathbf{a}), \qquad |\varepsilon_{\mathrm{lin}}(\mathbf{s}, \mathbf{a})| \le \epsilon_{\mathrm{lin}}(\mathbf{s}) \tag{22}$$

for all actions $\mathbf{a}$ on this segment. When $\epsilon_{\mathrm{lin}}(\mathbf{s}) = 0$, the margin is exactly affine along the interpolation direction.

**Assumption A.2.** Conditioned on $\mathbf{s}$, the nominal action $\mathbf{a}^\theta$ is sub-Gaussian: for any $\mathbf{u} \in \mathbb{R}^m$, $\mathbf{u}^\top(\mathbf{a}^\theta - \mathbb{E}[\mathbf{a}^\theta])$ is sub-Gaussian with variance proxy $\mathbf{u}^\top \Sigma_\theta(\mathbf{s})\mathbf{u}$. This holds exactly when $\mathbf{a}^\theta \sim \mathcal{N}(\mu_\theta(\mathbf{s}), \Sigma_\theta(\mathbf{s}))$.

**Assumption A.3.** The safe prior has nonnegative relative safety margin at $\mathbf{s}$:

$$\Delta_{\mathrm{safe}}(\mathbf{s}) = \hat{\Delta}(\mathbf{s}, \mathbf{a}^{\mathrm{safe}}(\mathbf{s})) \geq 0. \tag{23}$$

When strict reserve is required, we assume $\Delta_{\mathrm{safe}}(\mathbf{s}) > 0$.

## A.2 Useful Facts

**Definition A.4** (Sub-Gaussian random variable). A scalar random variable $X$ is $\sigma^2$-sub-Gaussian if

$$\mathbb{E}\left[\exp(t(X - \mathbb{E}[X]))\right] \leq \exp\left(\frac{t^2\sigma^2}{2}\right), \qquad \forall t \in \mathbb{R}. \tag{24}$$

**Lemma A.5.** *If $X$ is $\sigma^2$-sub-Gaussian, then for any $t > 0$,*

$$\Pr(X - \mathbb{E}[X] \leq -t) \leq \exp\left(-\frac{t^2}{2\sigma^2}\right). \tag{25}$$

*Proof.* This is the standard one-sided sub-Gaussian Chernoff bound; see Vershynin (2018). $\square$

**Corollary A.6.** *Let $X$ be $\sigma^2$-sub-Gaussian. Then*

$$\Pr(X < 0) \leq \delta \quad \Leftarrow \quad \mathbb{E}[X] \geq \sqrt{2\log\frac{1}{\delta}}\,\sigma. \tag{26}$$

*Equivalently, $\mathbb{E}[X] - \beta(\delta)\sigma \geq 0$ with $\beta(\delta) = \sqrt{2\log(1/\delta)}$.*

*Proof.* Apply Lemma A.5 with $t = \mathbb{E}[X]$. $\square$

**Corollary A.7** (Gaussian quantile). *If $X \sim \mathcal{N}(\mu, \sigma^2)$, then*

$$\Pr(X < 0) \leq \delta \quad \Longleftrightarrow \quad \mu \geq z_{1-\delta}\sigma, \tag{27}$$

*where $z_{1-\delta} = \Phi^{-1}(1 - \delta)$ is the $(1 - \delta)$ quantile of the standard normal distribution.*

## A.3 Closed-Form Intervention Weight

Under Assumption A.1 with $\epsilon_{\mathrm{lin}}(\mathbf{s}) = 0$, the relative margin is exactly affine along the interpolation segment. Therefore,

$$Z(\lambda) = (1 - \lambda)Z(0) + \lambda\Delta_{\mathrm{safe}}(\mathbf{s}). \tag{28}$$

Taking moments gives

$$\mathbb{E}[Z(\lambda)] = (1 - \lambda)\mu_\Delta(\mathbf{s}) + \lambda\Delta_{\mathrm{safe}}(\mathbf{s}), \qquad \mathrm{Var}(Z(\lambda)) = (1 - \lambda)^2\sigma_\Delta^2(\mathbf{s}). \tag{29}$$

**Proposition A.8.** *Assume the sufficient tail condition*

$$\mathbb{E}[Z(\lambda)] - \beta\sqrt{\mathrm{Var}(Z(\lambda))} \geq 0 \tag{30}$$

*for some $\beta > 0$. Under the exact interpolation identities in Eq. 29, define the risk-adjusted nominal margin*

$$L_\theta(\mathbf{s}) := \mu_\Delta(\mathbf{s}) - \beta\sigma_\Delta(\mathbf{s}). \tag{31}$$

*Then the smallest $\lambda \in [0, 1]$ satisfying Eq. 30 is*

$$\lambda^*(\mathbf{s}) = \mathrm{clip}_{[0,1]}\left(\frac{\beta\sigma_\Delta(\mathbf{s}) - \mu_\Delta(\mathbf{s})}{\Delta_{\mathrm{safe}}(\mathbf{s}) - \mu_\Delta(\mathbf{s}) + \beta\sigma_\Delta(\mathbf{s})}\right) = \begin{cases} 0, & L_\theta(\mathbf{s}) \geq 0, \\ \dfrac{-L_\theta(\mathbf{s})}{\Delta_{\mathrm{safe}}(\mathbf{s}) - L_\theta(\mathbf{s})}, & L_\theta(\mathbf{s}) < 0. \end{cases} \tag{32}$$

*This is Eq. 9 in the main text with $\beta = \beta(\delta(\mathbf{s}))$.*

*Proof.* Using Eq. 29,

$$\mathbb{E}[Z(\lambda)] - \beta\sqrt{\mathrm{Var}(Z(\lambda))} = (1-\lambda)\mu_\Delta + \lambda\Delta_{\mathrm{safe}} - \beta(1-\lambda)\sigma_\Delta \tag{33}$$

$$= (1-\lambda)L_\theta + \lambda\Delta_{\mathrm{safe}}. \tag{34}$$

If $L_\theta \geq 0$, the condition holds at $\lambda = 0$, so $\lambda^* = 0$. If $L_\theta < 0$, solving

$$(1-\lambda)L_\theta + \lambda\Delta_{\mathrm{safe}} \geq 0$$

gives

$$\lambda \geq \frac{-L_\theta}{\Delta_{\mathrm{safe}} - L_\theta}.$$

Clamping to $[0, 1]$ yields Eq. 32. $\qquad\square$

The expression in Eq. 32 has a geometric interpretation: the intervention weight is the ratio between the nominal safety deficit, $\max(0, -L_\theta(\mathbf{s}))$, and the total safety improvement available by moving from the nominal policy toward the safe prior, $\Delta_{\mathrm{safe}}(\mathbf{s}) - L_\theta(\mathbf{s})$.

## A.4   Robustness to Local Approximation Error

The exact derivation above assumes that the safety margin is affine along the interpolation segment. We next state a conservative perturbation bound when the local affine approximation has bounded error.

**Proposition A.9.** *Under Assumption A.1, define the ideal interpolated margin*

$$Z_{\mathrm{int}}(\lambda) := (1-\lambda)Z(0) + \lambda\Delta_{\mathrm{safe}}(\mathbf{s}). \tag{35}$$

*Then*

$$|Z(\lambda) - Z_{\mathrm{int}}(\lambda)| \leq 2\epsilon_{\mathrm{lin}}(\mathbf{s}). \tag{36}$$

*Consequently,*

$$|\mathbb{E}[Z(\lambda)] - ((1-\lambda)\mu_\Delta + \lambda\Delta_{\mathrm{safe}})| \leq 2\epsilon_{\mathrm{lin}}(\mathbf{s}), \tag{37}$$

*and*

$$\sqrt{\mathrm{Var}(Z(\lambda))} \leq (1-\lambda)\sigma_\Delta(\mathbf{s}) + 2\epsilon_{\mathrm{lin}}(\mathbf{s}). \tag{38}$$

*Proof.* Let

$$\varepsilon_\lambda = \varepsilon_{\mathrm{lin}}(\mathbf{s}, \tilde{\mathbf{a}}_\lambda), \quad \varepsilon_0 = \varepsilon_{\mathrm{lin}}(\mathbf{s}, \mathbf{a}^\theta), \quad \varepsilon_{\mathrm{safe}} = \varepsilon_{\mathrm{lin}}(\mathbf{s}, \mathbf{a}^{\mathrm{safe}}).$$

By Assumption A.1,

$$Z(\lambda) - Z_{\mathrm{int}}(\lambda) = \varepsilon_\lambda - (1-\lambda)\varepsilon_0 - \lambda\varepsilon_{\mathrm{safe}}.$$

Since each error term has magnitude at most $\epsilon_{\mathrm{lin}}(\mathbf{s})$,

$$|Z(\lambda) - Z_{\mathrm{int}}(\lambda)| \leq \epsilon_{\mathrm{lin}} + (1-\lambda)\epsilon_{\mathrm{lin}} + \lambda\epsilon_{\mathrm{lin}} = 2\epsilon_{\mathrm{lin}}.$$

Taking expectations gives Eq. 37. For the standard deviation bound, write

$$Z(\lambda) = Z_{\mathrm{int}}(\lambda) + \xi_\lambda, \qquad |\xi_\lambda| \leq 2\epsilon_{\mathrm{lin}}.$$

By the triangle inequality for standard deviation,

$$\mathrm{Std}[Z(\lambda)] \leq \mathrm{Std}[Z_{\mathrm{int}}(\lambda)] + \mathrm{Std}[\xi_\lambda] \leq (1-\lambda)\sigma_\Delta + 2\epsilon_{\mathrm{lin}}.$$

$$\square$$

It follows that a conservative sufficient condition under local approximation error is

$$(1-\lambda)L_\theta(\mathbf{s}) + \lambda\Delta_{\mathrm{safe}}(\mathbf{s}) \geq 2(1+\beta)\epsilon_{\mathrm{lin}}(\mathbf{s}). \tag{39}$$

Indeed, Eq. 37 and Eq. 38 imply

$$\mathbb{E}[Z(\lambda)] - \beta\,\mathrm{Std}[Z(\lambda)] \geq (1-\lambda)L_\theta + \lambda\Delta_{\mathrm{safe}} - 2(1+\beta)\epsilon_{\mathrm{lin}}. \tag{40}$$

Thus, the same risk buffer $\beta\sigma_\Delta$ that controls stochastic tail risk also provides robustness to local certificate mismatch when it dominates the linearization error.

### A.5    Near-Determinism at the Boundary

The main text motivates near-boundary determinism by requiring the tail budget to become more conservative as the system approaches the safety boundary. The following result formalizes this limiting behavior for the closed-form template.

**Proposition A.10.** *Assume $\sigma_\Delta(\mathbf{s}) > 0$ and $\Delta_{\text{safe}}(\mathbf{s}) \in (0, \infty)$. Let the risk tolerance be*

$$\delta(\mathbf{s}) = \exp\big(-\rho(\tilde{\Delta}(\mathbf{s}))\big),$$

*where $\tilde{\Delta}(\mathbf{s}) \in [0,1]$ is a normalized margin with $\tilde{\Delta}(\mathbf{s}) = 0$ at the safety boundary, and $\rho(x) \to \infty$ as $x \downarrow 0$. If the sub-Gaussian quantile $\beta(\delta) = \sqrt{2 \log(1/\delta)}$ is used, then for any sequence of states with $\tilde{\Delta}(\mathbf{s}) \downarrow 0$,*

$$\beta(\delta(\mathbf{s})) \to \infty \quad \Longrightarrow \quad \lambda^*(\mathbf{s}) \to 1 \quad \Longrightarrow \quad \tilde{\Sigma}(\mathbf{s}) = (1 - \lambda^*(\mathbf{s}))^2 \Sigma_\theta(\mathbf{s}) \to 0. \tag{41}$$

*Proof.* Using Eq. 32, for sufficiently large $\beta$,

$$\lambda^*(\mathbf{s}) = \frac{\beta \sigma_\Delta - \mu_\Delta}{\Delta_{\text{safe}} - \mu_\Delta + \beta \sigma_\Delta}.$$

As $\beta \to \infty$, the dominant terms give

$$\lambda^*(\mathbf{s}) \to \frac{\beta \sigma_\Delta}{\Delta_{\text{safe}} + \beta \sigma_\Delta} = 1 - \frac{\Delta_{\text{safe}}}{\Delta_{\text{safe}} + \beta \sigma_\Delta} \to 1,$$

because $\sigma_\Delta > 0$ and $\Delta_{\text{safe}} < \infty$. The covariance claim then follows immediately from Eq. 16. $\square$

Proposition A.10 motivates the endpoint behavior of the implemented rate, but AutoSafe does not need to estimate the risk potential $\rho$ explicitly. Instead, Eq. 10 enforces the boundary behavior by construction: $\lambda(0, p) = 1$ and $\lambda(1, p) = 0$ for all $p > 0$. The learnable sharpness parameter $p$ controls the finite-margin intervention profile, while the covariance collapse at the boundary follows directly from the affine policy composition.

# B    Safety Design for Simplex

There exist many approaches for safe policy design. In this work, we adopt the procedure introduced in (Seto & Sha, 1999) to synthesize a robust safe controller and derive a certified safety envelope using an over-approximated system model. In this appendix, we summarize the general formulation and solution procedure. Detailed implementation and experiment-specific parameters are provided in the supplementary code.

Consider the nonlinear system

$$\dot{\mathbf{s}} = \underbrace{\mathbf{As} + \mathbf{Ba}}_{\text{known}} + \underbrace{\mathbf{f}(\mathbf{s}, \mathbf{a})}_{\text{unknown}}, \tag{42}$$

where $\mathbf{s} \in \mathbb{R}^n$ and $\mathbf{a} \in \mathbb{R}^m$ denote the state and action, respectively.

The objective is to design a robust safe controller that keeps the system within the safety set

$$\mathcal{S}_f := \{\mathbf{s} \in \mathbb{R}^n \mid \mathbf{A}_s \mathbf{s} \le \mathbf{b}_s\}, \tag{43}$$

under the action constraints

$$\mathcal{A}_f := \{\mathbf{a} \in \mathbb{R}^m \mid \mathbf{A}_a \mathbf{a} \le \mathbf{b}_a\}. \tag{44}$$

Here, $\mathbf{A}_s \in \mathbb{R}^{n_s \times n}$ and $\mathbf{b}_s \in \mathbb{R}^{n_s}$ define the state constraints, while $\mathbf{A}_a \in \mathbb{R}^{n_a \times m}$ and $\mathbf{b}_a \in \mathbb{R}^{n_a}$ define the action constraints.

**Definition B.1** (Safety Definition). Consider the safety set $\mathcal{S}_f$ Eq. 43. The system Eq. 42 is said to be safe, if given any $\mathbf{s}_t \in \mathcal{S}_f$, the $\mathbf{s}_{t+1} \in \mathcal{S}_f$ holds for any time $t \in \mathbb{N}$.

Directly designing a verifiable safe controller for the nonlinear system Eq. 42 is difficult due to the unknown nonlinear term $\mathbf{f}(\mathbf{s}, \mathbf{a})$. Instead, we consider the linear over-approximation (Sha, 2001)

$$\dot{\mathbf{s}} = \mathbf{As} + \mathbf{Ba}. \tag{45}$$

We parameterize the safe controller as a linear feedback policy

$$\mathbf{a} = \mathbf{Fs}, \tag{46}$$

where $\mathbf{F} \in \mathbb{R}^{m \times n}$ is the controller gain matrix. The resulting closed-loop dynamics become

$$\dot{\mathbf{s}} = \bar{\mathbf{A}}\mathbf{s}, \qquad \bar{\mathbf{A}} = \mathbf{A} + \mathbf{BF}. \tag{47}$$

To jointly enforce state and action constraints, we define the unified constraint matrix

$$\mathbf{D} = \begin{bmatrix} \mathbf{A}_s \operatorname{diag}(\mathbf{b}_s)^{-1} \\ \mathbf{A}_a \mathbf{F} \operatorname{diag}(\mathbf{b}_a)^{-1} \end{bmatrix}, \tag{48}$$

such that the constraints can be compactly written as

$$\mathbf{Ds} \le \mathbf{1}. \tag{49}$$

The objective is to find a controller gain $\mathbf{F}$ such that the closed-loop system is asymptotically stable and admits a certified forward-invariant subset of the safety set.

**Definition B.2** (Quadratic Stability (Seto & Sha, 1999)). The closed-loop system Eq. 47 is quadratically stable if there exists a positive definite matrix $\mathbf{P} \succ 0$ such that the quadratic Lyapunov function

$$V(\mathbf{s}) = \mathbf{s}^\top \mathbf{P} \mathbf{s} \tag{50}$$

has negative derivative along all trajectories of the system.

The derivative of the Lyapunov function is

$$\dot{V}(\mathbf{s}) = \mathbf{s}^\top \left( \bar{\mathbf{A}}^\top \mathbf{P} + \mathbf{P}\bar{\mathbf{A}} \right) \mathbf{s}. \tag{51}$$

Therefore, the closed-loop system is asymptotically stable if there exists $\mathbf{P} \succ 0$ satisfying

$$\bar{\mathbf{A}}^\top \mathbf{P} + \mathbf{P}\bar{\mathbf{A}} \preceq 0. \tag{52}$$

The corresponding certified stability region is the ellipsoid

$$\Omega \triangleq \left\{ \mathbf{s} \in \mathbb{R}^n \mid \mathbf{s}^\top \mathbf{P}\mathbf{s} \leq 1 \right\}. \tag{53}$$

To maximize the size of this certified region, we introduce the change of variables

$$\mathbf{Q} = \mathbf{P}^{-1}, \qquad \mathbf{R} = \mathbf{F}\mathbf{Q}, \tag{54}$$

and formulate the following semidefinite program:

$$\min_{\mathbf{Q} \succ 0, \mathbf{R}} \quad -\log\det(\mathbf{Q}) \tag{55}$$

$$\text{s.t.} \quad \mathbf{D}\mathbf{Q}\mathbf{D}^\top \preceq \mathbf{I}, \tag{56}$$

$$\begin{bmatrix} \alpha\mathbf{Q} & (\mathbf{A}\mathbf{Q} + \mathbf{B}\mathbf{R})^\top \\ \mathbf{A}\mathbf{Q} + \mathbf{B}\mathbf{R} & \mathbf{Q} \end{bmatrix} \succeq 0, \tag{57}$$

$$\begin{bmatrix} \mathbf{Q} & \mathbf{R}^\top \\ \mathbf{R} & \frac{1}{\beta}\mathbf{I}_m \end{bmatrix} \succeq 0. \tag{58}$$

Solving the SDP in Eq. 55–Eq. 58 yields $\mathbf{Q}$ and $\mathbf{R}$, from which the Lyapunov matrix and controller gain are recovered as

$$\mathbf{P} = \mathbf{Q}^{-1}, \qquad \mathbf{F} = \mathbf{R}\mathbf{Q}^{-1}. \tag{59}$$

The matrices $\mathbf{P}$ and $\mathbf{F}$, together with the system matrices $\mathbf{A}$ and $\mathbf{B}$ used in the experiments, are provided in the accompanying code.

## C   Experimental Details

### C.1   Algorithm Implementations

#### C.1.1   Soft actor-critic (SAC) Baseline

Our implementation of Soft Actor-Critic follows (Haarnoja et al., 2018) and (Fujimoto et al., 2018), with default parameters summarized in Table 3. These parameters are shared across all baseline algorithms unless specified otherwise.

Table 2: Parameter setting for Soft Actor-Critic (SAC)

| Hyperparameter | Value |
|---|---|
| Discount factor ($\gamma$) | 0.99 |
| Learning rate (actor, critic) | $3 \times 10^{-4}$ |
| Optimizer | Adam |
| Target smoothing coefficient ($\tau$) | 0.005 |
| Entropy coefficient | 0.1 |
| Target update interval | 1 |
| Activation function | ReLU |
| Training steps per environment step | 1 |
| Evaluation period (steps) | 10000 |
| Neural Network (MLP) | [256, 256] |
| Batch size | 128 |

Table 3: Task Specific Parameter Setting

| Hyperparameter | CartPole | Glucose | Quadrotor | Quadruped |
|---|---|---|---|---|
| Replay Memory size | $2 * 10^5$ | $1 * 10^5$ | $1 * 10^6$ | $5 * 10^6$ |
| Entropy coefficient | 0.1 | 0.2 | 0.005 | 0.005 |
| Total training steps | $2 * 10^5$ | $1 * 10^5$ | $1 * 10^6$ | $5 * 10^5$ |
| Maximum steps per episode | 500 | 200 | 1000 | 3000 |

#### C.1.2   AutoSafe

**AutoSafe** fuses a standard DRL policy $\pi_\theta$ with a safe model-based policy $\pi_{\text{safe}}$ through a weighted summation of their action outputs: $\tilde{\mathbf{a}} = (1 - \lambda(\mathbf{s})) \cdot \mathbf{a}^\theta(\mathbf{o}) + \lambda(\mathbf{s}) \cdot \mathbf{a}^{\text{safe}}(\mathbf{s})$. Here, we use a more general expression for the input of DRL, because DRL can work on a broader range of information, for instance, images and augmented state information that includes past states. The specific observation designs can be found in the implementation details of the applications. The DRL policy $\pi_\theta$ takes the full observation $\mathbf{o}_t = (\mathbf{s}_t, \mathbf{i}_t)$, which includes the state vector $\mathbf{s}_t$ and additional information $\mathbf{i}_t$ (e.g., images), and generates an action $\mathbf{a}_\theta \sim \pi_\theta(\cdot \mid \mathbf{o}_t)$. The safe policy $\pi_{\text{safe}}$ only observes the state vector and outputs a safe action $\mathbf{a}_{\text{safe}} = \mathbf{Fs}$.

The mixing coefficient $\lambda(\mathbf{s}) \in [0, 1]$ is determined by an exponential-like activation function that takes as input a non-negative safety margin $\Delta_t \in \mathbb{R}_{\geq 0}$, computed by a non-learnable Safety monitor function, together with a learnable sharpness parameter $p$. When the agent operates well inside the safety envelope, $\lambda(\mathbf{s})$ is small and the action is dominated by $\mathbf{a}_\theta$. As the risk $z$ increases near the boundary of the safety set $\mathcal{S}_c$, $\lambda(\mathbf{s})$ grows, shifting the action towards $\mathbf{a}_{\text{safe}}$. At the boundary, the agent relies solely on the safe policy to ensure constraint satisfaction. The sharpness parameter $p$ controls the sharpness of this transition. An overview of the entire architecture is shown in Fig. 8.

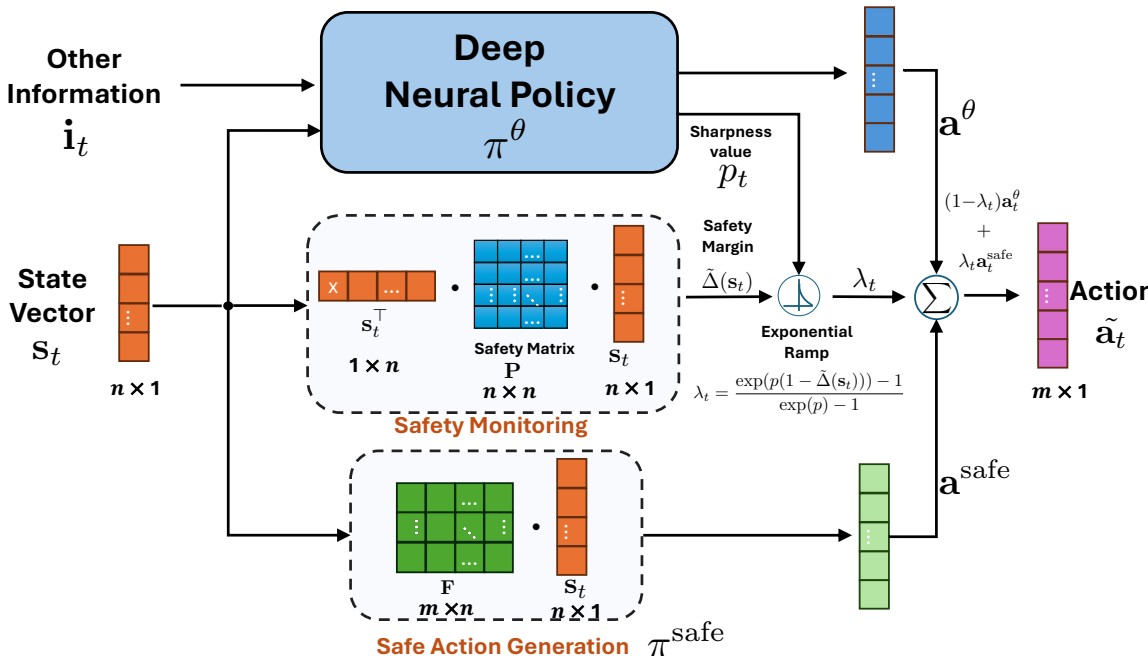

Figure 8: **AutoSafe**: The policy uses the state vector $\mathbf{s}_t$, included in the observation, to assess risk $\tilde{\Delta}(\mathbf{s}_t)$ and generate safe action $\mathbf{a}^{\text{safe}}$. In parallel, the learning-based component processes the full observation $\mathbf{o}_t = (\mathbf{s}_t, \mathbf{i}_t)$ to produce high-performance but potentially unsafe actions $\mathbf{a}^\theta$. The two action outputs are fused through a weighted summation, where the weights are determined by a learnable exponential activation function conditioned on the risk value.

The proposed **AutoSafe** neural policy additionally has a temperature prediction head compared to the standard actor network in SAC. This prediction head shares the same backbone as the action prediction head and is activated using the tanh activation function. We then apply an affine transformation to map the output range $(-1, 1)$ into $(p_{\min}, p_{\max})$. We set $p_{\min} = 1.0$ and $T_{\max} = 25.0$ for all studied case studies. The selection of the range can be flexible. We set $p_{\max} = 25.0$ to ensure the DRL agent has enough freedom to explore the state using its own action within the safety envelope. The value of $\lambda$ is capped to 1 for the case of exceeding the safety envelope caused by abrupt environmental uncertainties such as disturbance or noise.

### C.1.3 Simplex

The implementation of the Simplex architecture follows the standard setting as in (Sha, 2001; Phan et al., 2020). The system's safety is backed up by a safe policy. The switching between the safe and learning-based policy is determined by a state-dependent function, detailed as follows:

$$\tilde{\mathbf{a}}_t = \begin{cases} \mathbf{a}_t^{\text{safe}} = \pi^{\text{safe}}(\mathbf{s}_t), & \text{if } 1 - \mathbf{s}_t^\top \mathbf{P}\mathbf{s} \leq \Delta_{\min}, i.e, \ \mathbf{s} \notin \Omega \\ \mathbf{a}^\theta_t \sim \pi^\theta(\cdot \mid \mathbf{o}_t), & \text{otherwise.} \end{cases}$$

The matrix $\mathbf{P}$ is solved from the LMIs problem in B and identical to the $\mathbf{P}$ in **AutoSafe**. The safe policy $\pi_{safe}$ generates the safe action as $\mathbf{a}_{safe} = \mathbf{F}\mathbf{s}$.

### C.1.4 CBF

In this experiment, we implement a Control-Barrier Function based approach for the safety back up mechanism. Our implementation of CBF follows the setup introduced in (Ames et al., 2019). In particular, given an action from a DRL policy $\mathbf{a}_\theta$, a CBF-based safety filter computes a safe action $\mathbf{a}_{safe}$ by solving the

following quadratic program:

$$\mathbf{a}_{safe} = \arg\min_{\mathbf{a}} \frac{1}{2}\left\|\mathbf{a} - \mathbf{a}_\theta\right\|^2 \tag{60}$$

$$\text{s.t.} \quad L_f h(\mathbf{s}) + L_g h(\mathbf{s})\mathbf{a} \geq -\alpha(h(\mathbf{s})) \tag{61}$$

$$\mathbf{A}_a \mathbf{a} \leq \mathbf{b}_a, \tag{62}$$

where $\alpha(h(\mathbf{s})) = k \cdot h(\mathbf{s})$ and $h(\mathbf{s})$ is defined as:

$$h(\mathbf{s}) = \mathbf{b}_s - \mathbf{A}_s \mathbf{s} \quad h(\mathbf{s}) \in \mathbb{R}^{n_s}. \tag{63}$$

Here, $h(\mathbf{s})$ is defined in matrix form, yielding a vector-valued barrier function. In practice, each component of this vector is enforced as an individual CBF constraint. The $(\mathbf{A}_s, \mathbf{b}_s)$ and $(\mathbf{A}_a, \mathbf{b}_a)$ correspond to the state constraints and action constraints, respectively, as introduced in the background section. For all experiments, we set $k = 0.5$.

The results are shown in Figure 10. We observed that the constructed CBF baseline reduces safety violations, but its performance remains below that of the Simplex approach. In practice, we found that solving the CBF QP online is sensitive to parameter choices and can frequently become infeasible, especially near the safety boundary. Additional parameter tuning and system-specific tuning may improve its performance, but this process is nontrivial and beyond the scope of this work.

## C.2 Applications Details

### C.2.1 Cartpole

**Task Definition:** The goal of this task is to balance the pole at the intended target position $\hat{x}$ by driving the cart using force input. In this task, the observation of the agent is defined as $o_t = \{x_t, \dot{x}_t, \sin(\theta), \cos(\theta), \dot{\theta}\}$. Our method and the other methods that use a model-based safe prior require the tracking error $e_t$ as the additional input for the safe policy to generate a safe action. $e_t$ is defined as the difference between the current state $\mathbf{s}_t$ and the control equilibrium $\mathbf{s}^*$ of the model-based design, $e_t = \{x_t - x^*, \dot{x} - \dot{x}^*, \theta_t - \theta^*, \dot{\theta}_t - \dot{\theta}^*\}$. The equilibrium is set as $\mathbf{s}^* = \{0, 0, 0, 0\}$. The control loop is running at 50Hz.

We append the tracking error $e_t$ to the observation space of the other model-free methods to ensure all algorithms receive the same amount of information. We adopt the reward function proposed in (Cao et al., 2022), formulated as

$$r = e^{-\delta \cdot d(\mathbf{s}_t - \hat{\mathbf{s}})} - \beta \cdot \mathbf{a}_t^2,$$

where $\delta = 5$ is a parameter that adjusts the smoothness of the exponential function; $d(\cdot)$ is the Euclidean distance between the current state $\mathbf{s}_t$ and the control target $\hat{\mathbf{s}} = \{0.1, 0, 0, 0\}$; $\beta$ is a parameter to balance the reward and the action penalty. For more details about the task setting, we refer interested readers to (Cao et al., 2022).

**Safety Constraints:** In this task, the safety constraints are

$$|x_t| \leq x_{\lim} \quad |\theta_t| \leq \theta_{\lim},$$

where $x_{\lim} = 0.5$ m and $\theta_{\lim} = 0.785$ rad.

**Safety Design** The safety envelope and safe policy are obtained from solving an LMI problem, as discussed in Section B. The system model can be found at (Florian, 2007). The linearized model and the code to calculate the matrix $\mathbf{P}$ and $\mathbf{F}$ are available in the attached supplementary files. For more details of solving LMIs for the cartpole task, we refer the interested reader to (Cao et al.; Seto & Sha, 1999).

### C.2.2 Glucose

**Task Definition** The problem setting and simulation of this application are adopted from (Tian et al., 2024). In the blood glucose regulation task, the goal is to regulate the blood glucose level $G$ to minimize

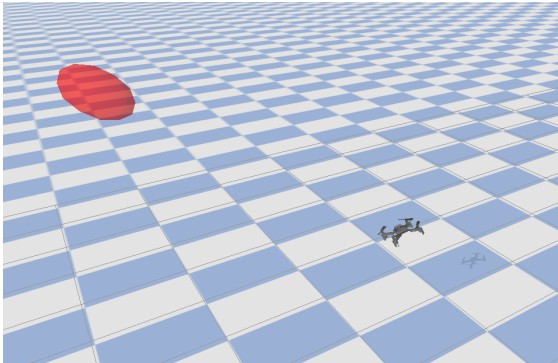 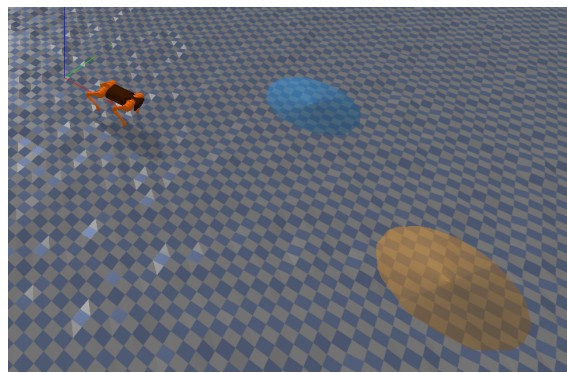

(a) Setup of quadrotor goal reaching. The red sphere represents the target zone.

(b) Setup of quadruped navigation. The target pick-up zone is indicated as a blue spot, and the delivery zone is indicated as a yellow spot. The terrain is randomized with uneven height.

Figure 9: Experimental setup illustration of quadrotor and quadruped robots

the Magni risk (Fox et al., 2020) by controlling insulin injection $a_I$. The dynamics of the glucose control problem are governed by the following ODEs (Tian et al., 2024),

$$\dot{G} = -p_1(G - G_b) - GX + D_t,$$
$$\dot{X} = -p_2 X + p_3(I - I_b),$$
$$\dot{I} = -n(I - I_b) + a_I$$

Here, $G$ represents the amount of glucose in the blood, and $I$ represents the amount of insulin in the blood. $X$ describes the delayed effect of insulin on lowering blood glucose, which is often unobservable. In this task, the observation is the $o_t = \{G_t, \Delta G_t, t\}$, where $\Delta G_t = G_t - G_{t-1}$ and $t$ is the total time passed after meal ingestion. The equilibrium of the model-based design is set to be the normal fasting level of glucose and insulin. $\mathbf{s}^* = \{G^*, X^*, I^*\} = \{138, 0, 7\}$. The reward function is defined as

$$r = \begin{cases} -(3.35506 \times ((\ln^{0.8353}) - 3.7932)^2 & \text{if } 10 \leq G \leq 1000, \\ -1e3 & \text{otherwise .} \end{cases}$$

**Safety Constraints** We follow the safety constraints introduced in (Tian et al., 2024) as:

$$G_{\min} \leq G_t \leq G_{\max},$$

where $G_{\min} = 10$ and $G_{\max} = 1000$.

**Moded-based Design** The safety envelope and safe policy are obtained from solving an LMI problem, as discussed in Section B. The system model can be found at (Tian et al., 2024). The linearized model and the code to calculate the matrix $\mathbf{P}$ and $\mathbf{K}$ are available in the attached supplementary files.

### C.2.3  3D Quadrotor Goal Reaching

**Task Definition** The goal of this task is to control a quadrotor to reach a target goal position $\{\hat{x}, \hat{y}, \hat{z}\}$ by controlling the thrust input on each propeller. The observation for the agent is $o = \{x, y, z, \theta_x, \theta_y, \theta_z, \dot{x}, \dot{y}, \dot{z}, \dot{\theta}_x, \dot{\theta}_y, \dot{\theta}_z\}$. The action space consists of a four-dimensional trust input, denoted as $a = \{u_1, u_2, u_3, u_4\}$. The equilibrium of the model-based design is $s^* = \{0, 0, 0, 0, 0, 0, 0, 0, 0, 0, 0, 0\}$. The reward function is adopted from (Yuan et al., 2022) as:

$$r = e^{-\alpha \cdot (\|x-\hat{x}\|^2 + \|y-\hat{y}\|^2 + \|z-\hat{z}\|^2) - \beta \cdot \|a\|^2},$$

where $\alpha = 1.0$ and $\beta = 1e-4$ are the weights to balance the distance-related reward and action penalty. In our case study, we set the initial position of the quadrotor as $\mathbf{s}_{0_{xyz}} = \{1.5, 1.5, 1.5\}$ and the target position of the quadrotor as $\hat{\mathbf{s}}_{xyz} = \{2.5, 2.5, 2.5\}$. The control loop is running at 50Hz.

**Safety Constraints** The safety constraint is defined as

$$x_{\min} < x_t < x_{\max}, \quad y_{\min} < y_t < y_{\max}, \quad z_{\min} < z_t < z_{\max} \tag{64}$$

where $x_{\min} = -5.0$ m, $y_{\min} = -5.0$ m, $z_{\min} = 0.0$ m, and $x_{\max} = y_{\max} = z_{\max} = 5.0$ m, representing the allowable moving area in the $x - y - z$ space.

**Safety Design** The safety envelope and safe policy are obtained from solving an LMI problem, as discussed in Section B. The system model can be found at (Yuan et al., 2022). The linearized model and the code to calculate the matrix $\mathbf{P}$ and $\mathbf{F}$ are available in the attached supplementary files.

### C.2.4 Quadruped Navigation on Uneven Terrain

**Task Definition** In this task, we aim to train a safe RL policy to enable the quadruped robot to walk through the uneven terrain to finish a virtual package delivery problem. The robot needs to first go to a package pick-up zone (A) and then navigate a package drop zone (B). We assume a virtual package is automatically attached to the robot when the robot reaches zone A and detached when it reaches zone B. The terrain is created unevenly by randomly placing blocks on the floor, with the maximum height of the unevenness to be 4 centimeters. The control loop is running at 200Hz.

The observation of the agent includes the pose of the robot in the world coordinates and the relative distance toward goal A for picking up, and the relative distance toward goal B for dropping off. We additionally add the task phase ID, $i$ (0 or 1), for the picking up and delivery. Overall, $o$ is defined as $o = \{x, y, z, \theta_x, \theta_y, \theta_z, \dot{x}, \dot{y}, \dot{z}, \dot{\theta}_x, \dot{\theta}_y, \dot{\theta}_z, x_{rel}, y_{rel}, z_{rel}, i\}$. The action space is defined as the desired acceleration as $a = \{\ddot{x}, \ddot{y}, \ddot{z}, \ddot{\theta}_x, \ddot{\theta}_y, \ddot{\theta}_z\}$. The acceleration input is then mapped to the low-level joint angles using a Model Predictive Controller (MPC) (Yang et al., 2022b). The equilibrium point for the model-based design is defined as $s^* = \{0, 0, 0.24, 0, 0, 0, 0.26, 0, 0, 0, 0, 0, 0\}$, meaning maintaining the height $z = 0.24$ and forward velocity $\dot{x} = 0.26$ m/s. The reward is designed piece-wise to incentivise the robot to move to A and B to finish the whole task, as

$$r(\mathbf{s}) = w_d \, r_d(\mathbf{s}, g) + w_h \, r_h(\mathbf{s}, g) + w_z \, r_z(\mathbf{s}) + R_1 \, \mathbb{I}_{\text{pickup}} + R_2 \, \mathbb{I}_{\text{delivery}}.$$

Each term of the reward is defined as:

$$\text{positional reward} : r_d(\mathbf{s}, g) = -\|\mathbf{s}_{xyz} - \hat{\mathbf{s}}_{xyz}\|,$$
$$\text{directional reward} : r_h(\mathbf{s}, g) = -|\theta - \theta_g|,$$
$$\text{height regulation reward} : r_z(\mathbf{s}) = -|z - z_{\text{ref}}|,$$
$$\mathbb{I}_{\text{pickup}} = \begin{cases} 1, & \text{if reaches the pickup zone (A)} \\ 0, & \text{otherwise} \end{cases}$$
$$\mathbb{I}_{\text{delivery}} = \begin{cases} 1, & \text{if reaches the delivery zone (B)} \\ 0, & \text{otherwise} \end{cases}$$

where the coefficient of each term is detailed as

$$w_d = 1.0, \quad w_h = 0.25, \quad w_z = 0.25 \quad R_1 = 2.5, \quad R_2 = 50.$$

**Safety Constraints** The safety constraint in this task is mainly considered as the robot not falling and jumping too high while moving forward, as:

$$z_{\min} \le z_t \le z_{\max}$$

where $z_{\min} = 0.16$ m and $z_{\max} = 0.8$ m.

**Safety design** The safety envelope and safe policy are obtained from solving an LMI problem, as discussed in Section B. The linearized model and the code to calculate the matrix $\mathbf{P}$ and $\mathbf{F}$ are available in the attached supplementary files.

# D   Additional Experimental Results

In this section, we summarize the additional experimental results to provide more insights.

## D.1   Experimental results for all methods on all considered experiments.

We summarize the training progress and the accumulated safety violations in Fig 10. The $\Delta_{\min} = 0$ is used for Simplex and AutoSafe as the default value. The detailed ablation study on $\Delta_{\min}$ is shown in Table 4.

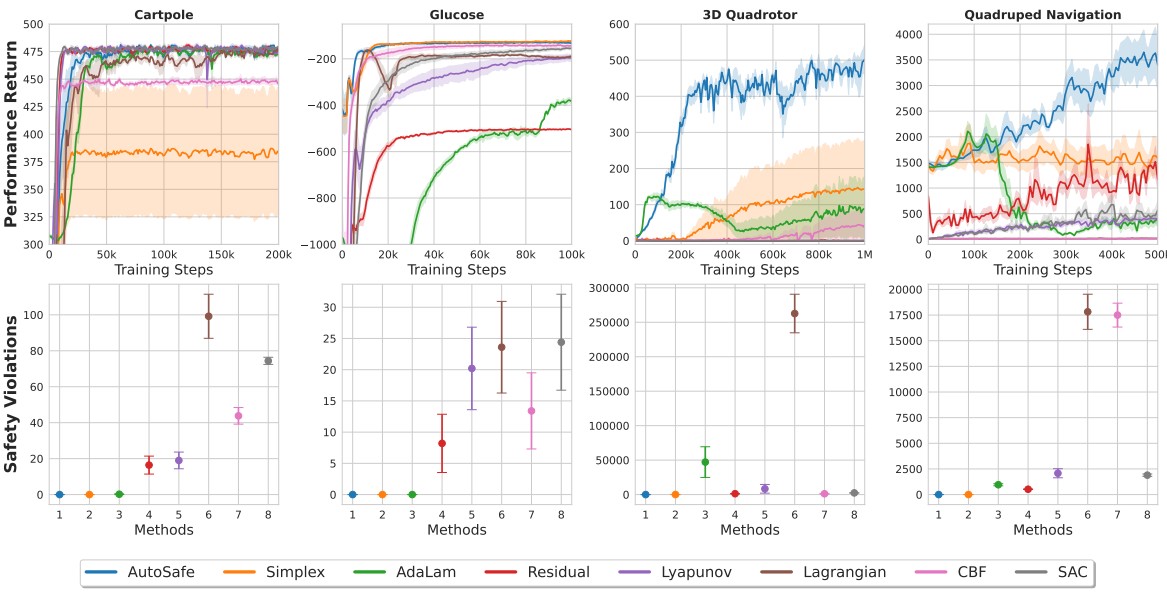

Figure 10: Performance return curve and the accumulated safety violation for all algorithms in the considered case studies

## D.2   Divergence of the hard-safety intervention

We visualize the training loss curves for *AutoSafe* and *Simplex*. We observe that training of the *Simplex* is gradually diverging with a large critic loss, as shown in Fig. 11. We attribute this to the frequent interventions by the safe policy, which creates discontinuities in the data distribution, meaning the state–action transitions suddenly shift from those produced by the learning policy to those imposed by the safe controller. This mismatch makes the data less smooth and harder for the neural network to approximate, causing unstable or diverging learning behavior.

## D.3   Ablation on the setting of Temperature $p$

In this case study, we compare our learnable temperature setting against two heuristics, including linear and exponential increasing. The result is shown in Fig. 13. We found that all methods work similarly regarding performance, except for the quadrotor case. However, designing an effective scheduling scheme is non-trivial, which requires repetitive manual tuning. Moreover, the manual setting might introduce bias. We visualize the evolution of the learnable sharpness parameter $T$ during the policy learning in Fig. 12. It can be seen that the learned sharpness converges to different values across tasks, suggesting that it may not be trivial to manually set the "right" parameter using heuristics.

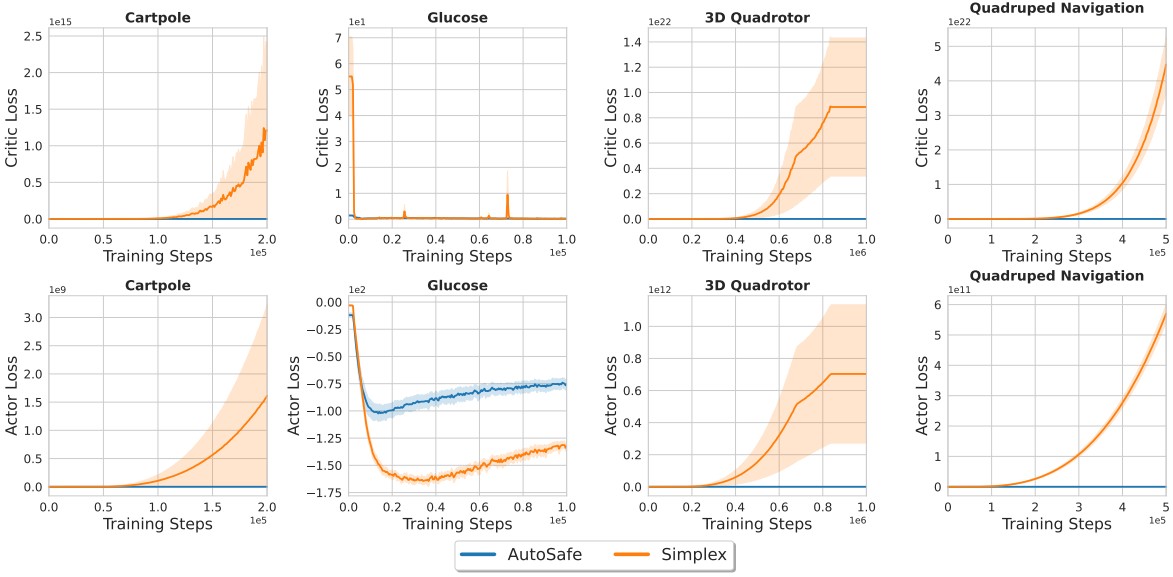

Figure 11: Critic Loss and Actor Loss of AutoSafe and Simplex-Based Method

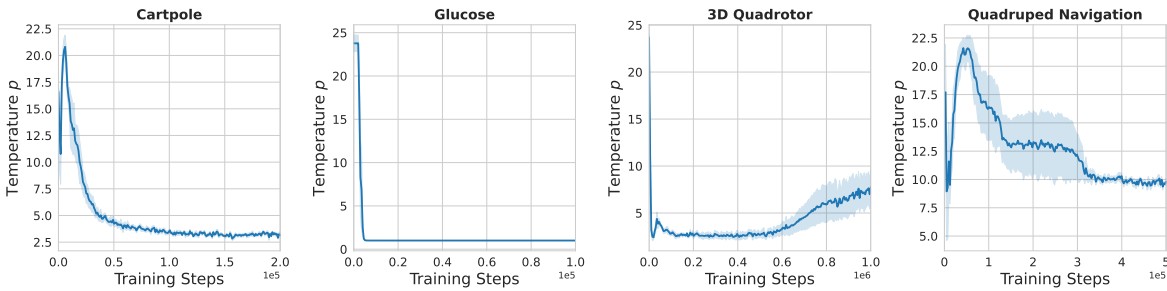

Figure 12: The evolution of sharpness parameter $p$ during policy learning for studied applications

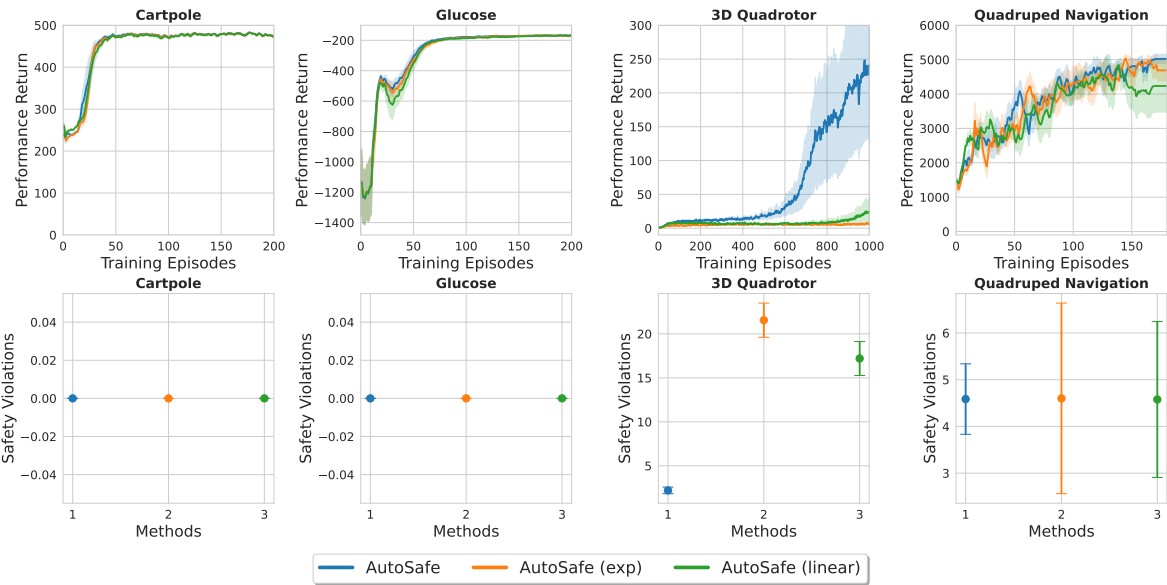

Figure 13: Ablation Study of sharpness parameter $p$: Learning-based vs. Schedule-based.

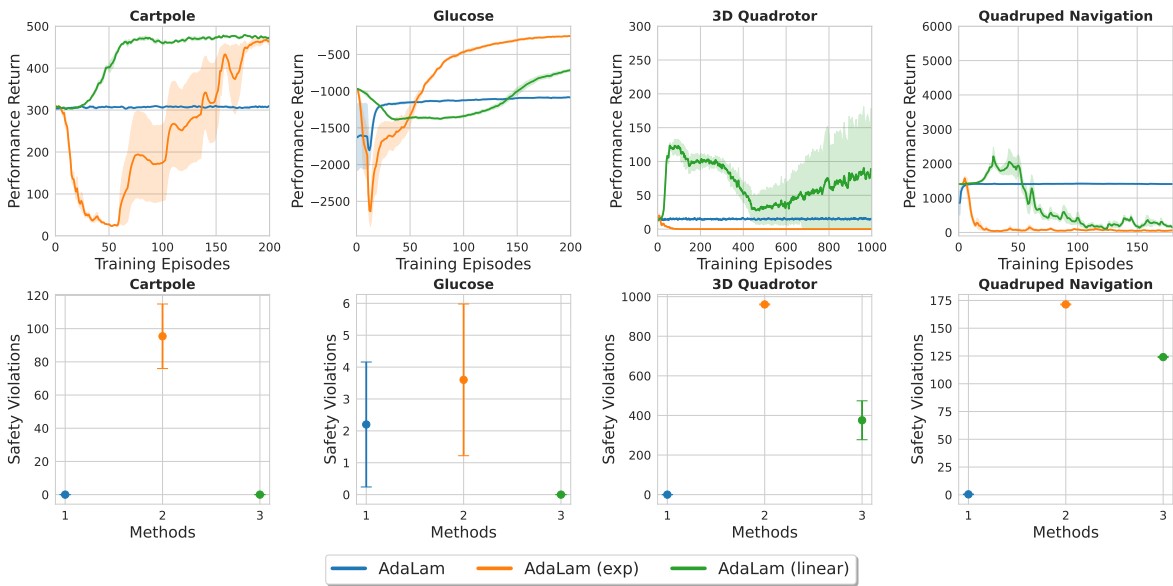

Figure 14: Ablation of the adaptation for $\lambda$ in AdaLam

## D.4    Ablation of AdaLam

In this study, we investigate several ways of setting the weights between safe action and learning-based action, as shown in Fig. 14. We found out that initializing the $\lambda$ to close to 1 at the beginning of the training enables safe interactions. For simple tasks, such as cartpole and glucose, the agent could learn using the data generated by the safe policy. However, we found that it is not effective in high-dimensional cases. For a learning-based setting, the exploration is not effective; therefore, the performance of the policy is barely improved. For the scheduled setting, we found that frequent safety violations occur when $\lambda$ decreases. The frequent safety violations generate a lot of uninformative data, where the agent cannot learn to converge.

## D.5    Under varying constraints

In this section, we demonstrate that the current safety design can be readily extended to handle online constraints with slight modification. Specifically, the offline recovery region can be reinterpreted and transformed into a closed-form expression to accommodate new constraints while preserving local invariance guarantees.

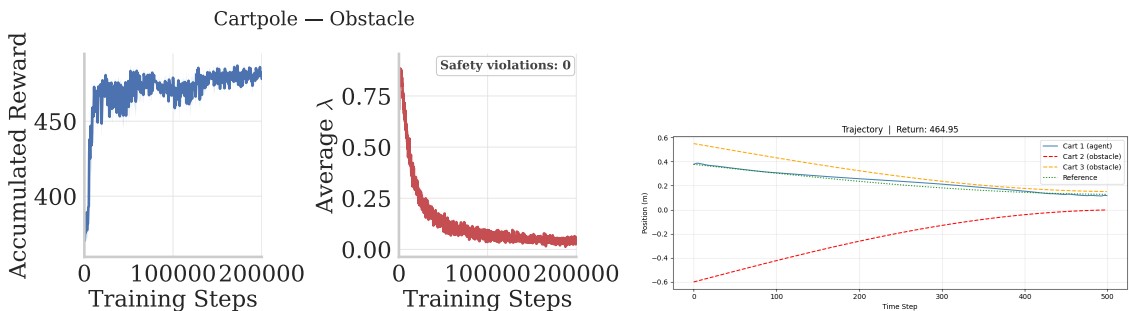

Figure 15: State trajectories over an evaluation episode. The dashed lines indicate the time-varying constraints on the position imposed by each intruder.

**CartPole**   The DRL agent learns to drive the cart-pole system toward a time-varying target position $x_t^*$ while two moving obstacle constraints dynamically shrink the admissible state space. The state space is $\mathcal{S} \subset \mathbb{R}^4$, and the action space is $\mathcal{A} \subset \mathbb{R}^1$.

The system is subject to the following fixed safety constraints:

$$|\theta_t| \le \theta_{\max}, \qquad |x_t| \le x_{\max}. \tag{65}$$

In addition, two intruder-induced time-varying constraints impose dynamic bounds on the cart position:

$$x_t \ge \ell_t, \qquad x_t \le r_t, \tag{66}$$

where $\ell_t$ and $r_t$ are updated online according to the motion of the intruders.

As shown in Fig. 15, **AutoSafe** learns to safely drive the cartpole to reach the target goal under two moving objects constraints without safety violations.

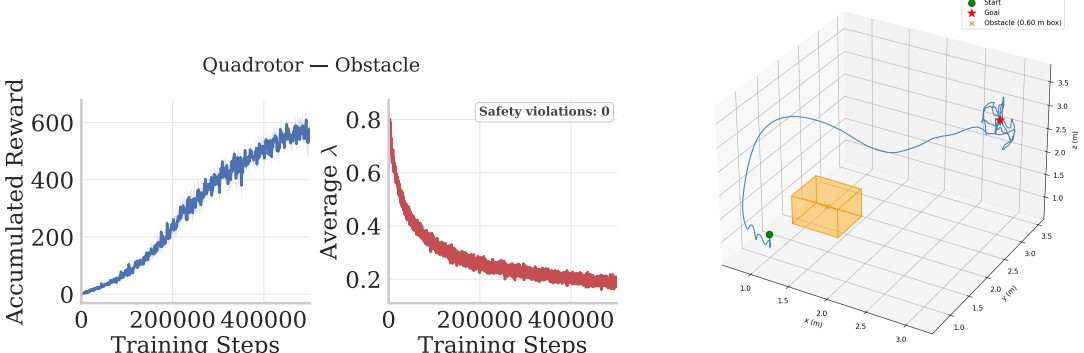

Figure 16: example evaluation trajectories with randomized obstacle positions. The adapted safe recoverable region enables safe clearance for goal reaching.

**Quadrotor**   A quadrotor must reach a goal position in 3D space while avoiding a static obstacle. The state space is $\mathcal{S} \subset \mathbb{R}^{12}$, and the action space is $\mathcal{A} \subset \mathbb{R}^4$.

The system is subject to the following fixed safety constraints:

$$|\phi_t| \le \phi_{\max}, \qquad |\psi_t| \le \psi_{\max}, \qquad |\vartheta_t| \le \vartheta_{\max}, \tag{67}$$

and an altitude constraint

$$z_t \ge z_{\min}. \tag{68}$$

In addition, the obstacle is represented by a collection of time-varying supporting hyperplane constraints:

$$a_{i,t}^\top p_t \ge b_{i,t}, \qquad i = 1, \dots, F, \tag{69}$$

where each obstacle face $i$ defines a supporting half-space that is updated online according to the relative motion of the quadrotor.

As shown in Fig. 16, *AutoSafe* learns to safely drive the quadrotor to reach the target goal while avoiding the obstacle.

Table 4: Effect of $\Delta_{\min}$ on performance and safety across four continuous control tasks. Mean ± std over five random seeds.

| Method | Cartpole | | Glucose | | Quadrotor | | Quadruped | |
|---|---|---|---|---|---|---|---|---|
| Method | Return ↑ | Viol. ↓ | Return ↑ | Viol. ↓ | Return ↑ | Viol. ↓ | Return ↑ | Viol. ↓ |
| AutoSafe$_{\Delta_{\min}=0.0}$ | 477.7 ± 2.5 | 0.0 ± 0.0 | -131.2 ± 3.7 | 0.0 ± 0.0 | 498.0 ± 87.1 | 1.0 ± 1.2 | 3484.7 ± 985.7 | 1.0 ± 1.2 |
| SimplexRL$_{\Delta_{\min}=0.0}$ | 384.5 ± 127.9 | 0.0 ± 0.0 | -124.0 ± 4.1 | 0.0 ± 0.0 | 142.9 ± 297.0 | 53.6 ± 61.1 | 1558.6 ± 690.0 | 1.4 ± 2.1 |
| AutoSafe$_{\Delta_{\min}=0.25}$ | 476.6 ± 3.3 | 0.0 ± 0.0 | -133.0 ± 4.4 | 0.0 ± 0.0 | 564.0 ± 68.1 | 0.2 ± 0.4 | 2549.6 ± 971.9 | 0.6 ± 0.9 |
| SimplexRL$_{\Delta_{\min}=0.25}$ | 395.8 ± 111.8 | 0.0 ± 0.0 | -126.2 ± 3.0 | 0.0 ± 0.0 | 13.9 ± 11.5 | 30.4 ± 48.5 | 2072.8 ± 758.0 | 0.6 ± 0.9 |
| AutoSafe$_{\Delta_{\min}=0.5}$ | 475.2 ± 2.4 | 0.0 ± 0.0 | -134.1 ± 3.3 | 0.0 ± 0.0 | 740.8 ± 27.2 | 0.0 ± 0.0 | 2015.2 ± 757.7 | 0.0 ± 0.0 |
| SimplexRL$_{\Delta_{\min}=0.5}$ | 419.3 ± 75.3 | 0.0 ± 0.0 | -126.1 ± 1.8 | 0.0 ± 0.0 | 17.5 ± 10.7 | 50.0 ± 53.7 | 1880.5 ± 461.8 | 0.2 ± 0.4 |
| AutoSafe$_{\Delta_{\min}=0.75}$ | 469.4 ± 8.1 | 0.0 ± 0.0 | -139.7 ± 5.9 | 0.0 ± 0.0 | 693.8 ± 66.5 | 0.0 ± 0.0 | 1479.2 ± 76.9 | 0.0 ± 0.0 |
| SimplexRL$_{\Delta_{\min}=0.75}$ | 430.5 ± 2.4 | 0.0 ± 0.0 | -129.2 ± 2.8 | 0.0 ± 0.0 | 22.7 ± 5.7 | 0.0 ± 0.0 | 1969.6 ± 427.4 | 0.0 ± 0.0 |
| AutoSafe$_{\Delta_{\min}=1.0}$ | 304.3 ± 1.4 | 0.0 ± 0.0 | -141.5 ± 13.1 | 0.0 ± 0.0 | 9.9 ± 0.0 | 0.0 ± 0.0 | 1416.6 ± 34.7 | 0.0 ± 0.0 |
| SimplexRL$_{\Delta_{\min}=1.0}$ | 304.3 ± 1.4 | 0.0 ± 0.0 | -129.7 ± 2.8 | 0.0 ± 0.0 | 11.2 ± 0.0 | 0.0 ± 0.0 | 1416.6 ± 34.6 | 0.0 ± 0.0 |

## D.6 Sensitivity Analysis on $\Delta_{\min}$

In Table 4, $\Delta_{\min} = 0$ corresponds to the most aggressive setting, where the safety switch happens when the system exits the maximum safety recoverable region. $\Delta_{\min} = 1.0$ corresponds to the most conservative setting, where $\lambda = 1$, meaning the system is fully controlled by the $\pi^{\text{safe}}$. As expected, larger $\Delta_{\min}$ leads to fewer safety violations due to tighter safety enforcement, but may also reduce task performance by constraining exploration. *AutoSafe* outperforms *Simplex* in most tasks, while showing comparable performance on the Glucose Regulation benchmark. As $\Delta_{\min} \to 1$, both methods converge toward the same behavior.

## D.7 Walltime comparison

Table 5: **Computation Time Comparison.** Average wall-clock time (in milliseconds) per step for inference and optimization steps. Mean ± std computed over all logged episodes (one seed). Lower is better (↓). Most time-consuming method per column is underlined.

| | Cartpole | | Glucose | | 3D Quadrotor | | Quadruped | |
|---|---|---|---|---|---|---|---|---|
| Method | Inf.↓ | Opt.↓ | Inf.↓ | Opt.↓ | Inf.↓ | Opt.↓ | Inf.↓ | Opt.↓ |
| SAC (Haarnoja et al., 2018) | 2.55 ± 0.34 | 3.19 ± 0.54 | 4.38 ± 0.55 | 5.76 ± 0.31 | 2.70 ± 0.28 | 3.20 ± 0.33 | 2.34 ± 0.21 | 3.34 ± 0.23 |
| SimplexRL (Phan et al., 2020) | 2.42 ± 0.73 | 3.36 ± 0.54 | 2.88 ± 0.85 | 5.68 ± 0.45 | 0.31 ± 0.01 | 2.87 ± 0.09 | 0.65 ± 0.07 | 3.08 ± 0.32 |
| CBF (Ames et al., 2019) | 8.16 ± 2.29 | 3.45 ± 1.25 | 8.84 ± 2.46 | 5.06 ± 1.63 | 8.00 ± 1.15 | 3.57 ± 0.61 | 8.86 ± 1.42 | 3.20 ± 0.54 |
| AdaLam (Tian et al., 2024) | 3.68 ± 0.92 | 4.22 ± 1.13 | 6.44 ± 1.50 | 6.80 ± 1.79 | 3.56 ± 0.54 | 4.06 ± 0.58 | 3.76 ± 0.01 | 5.06 ± 0.01 |
| Residual (Johannink et al., 2019) | 2.75 ± 0.36 | 3.34 ± 0.52 | 4.49 ± 0.51 | 5.83 ± 0.29 | 2.81 ± 0.27 | 3.24 ± 0.35 | 2.39 ± 0.20 | 3.32 ± 0.30 |
| Lyapunov (Westenbroek et al., 2022) | 2.66 ± 0.35 | 3.35 ± 0.52 | 4.40 ± 0.55 | 5.76 ± 0.31 | 2.94 ± 0.38 | 3.32 ± 0.41 | 2.36 ± 0.23 | 3.36 ± 0.23 |
| Lagrangian (Ha et al., 2020) | 2.60 ± 0.37 | 3.72 ± 0.64 | 4.25 ± 0.64 | 6.53 ± 0.78 | 2.53 ± 0.56 | 3.21 ± 0.71 | 2.37 ± 0.20 | 3.68 ± 0.28 |
| **AutoSafe (Ours)** | 3.68 ± 0.88 | 3.73 ± 0.89 | 4.30 ± 0.45 | 5.87 ± 0.27 | 3.60 ± 0.47 | 3.58 ± 0.43 | 2.82 ± 0.58 | 3.19 ± 0.70 |

In this experiment, we observe that CBF (optimization-based) incurs the highest inference cost due to the per-step online optimization loop, and shows higher variance in action generation. AdaLam requires higher optimization time, as the mixing parameter is trained separately from the Q-network. We see that SimplexRL achieves a relatively faster inference time because when the safe policy is activated, the inference switches from network forward pass to cheap matrix multiplication. AutoSafe maintains low computational overhead for action inference and policy optimization comparable to the other baselines, while achieving strong performance in terms of return and safety violations.

