# OpenReview forum: "Safe Online Learning via Smooth Safety-Structured Policy Composition"
_TMLR — Under review for TMLR_

### Review · Reviewer_EkHr · 2026-06-11

**Summary Of Contributions:**

- This work proposes a differentiable policy composition framework AutoSafe, which realizes smooth risk-aware intervention and resolves the gradient discontinuity issue of conventional safety filters.
- It provides corresponding theoretical analysis on the mechanism, and comprehensive simulations plus real-world tests verify its effectiveness, stability and practicality across various continuous control tasks.

**Audience:**

Yes

**Audience Explanation:**

1. The proposed smooth policy composition effectively eliminates gradient discontinuities caused by hard switching in traditional safety filters, achieving stable online learning while maintaining strict safety guarantees.

2. Extensive simulations and real-world cart-pole experiments demonstrate superior task performance across low- and high-dimensional continuous control tasks, with competitive computational efficiency.

**Claims And Evidence:**

No

**Claims Explanation:**

1. General applicability vs. practical assumptions
The abstract and introduction present AutoSafe as a general solution for safe online reinforcement learning and real-world deployment. However, the method relies on strong prerequisites: accurate system linearization, offline construction of certified safe controllers and safety margins via LMI and Lyapunov tools, as well as convex action spaces and ellipsoidal safe sets. These critical assumptions are only briefly noted in the appendix and limitations, rather than clearly stated in the main text, which mismatches the claimed generality.

2. Gap between theoretical derivation and practical implementation
Section 4.3 derives a closed-form optimal intervention weight lambda*(s)
 as the theoretical foundation. Nonetheless, this analytical solution is not used in practice; an exponential function is adopted instead. The deviation between the implemented lambda and the theoretically optimal lambda*(s) is neither quantified nor discussed.

3. Insufficient experiments on dynamic constraints and real-world generalization
The claim that the method adapts to time-varying constraints and moving obstacles is not fully validated. Current tests only consider low-speed moving obstacles, while challenging cases such as rapidly switched boundaries and unmodeled obstacles are missing. The performance limit under varying constraint speeds is also unexplained, making the relevant conclusion overstated. Additionally, real-world experiments are limited to a single cart-pole platform, so the generalization across diverse physical systems remains unproven.

4. Supplementary Question (Not a direct weakness, but for discussion)
When the agent enters high-risk regions near safety boundaries, the policy gradient tends to approach zero and becomes nearly ineffective. It is worth discussing whether this design is reasonable. Specifically, will the near-zero gradient prevent the policy from being updated in the long run, trap the agent in high-risk areas, and eventually degrade the overall performance to a state fully relying on the offline safe controller?

**Requested Changes:**

See above

---

### Review · Reviewer_qaPN · 2026-07-02

**Summary Of Contributions:**

This paper proposes AutoSafe, a safe online reinforcement learning architecture that composes a learned policy with a certified safe prior through a smooth, state-dependent mixing weight.

**Strengths**
1. The motivation is strong. If a safety module frequently overwrites the learner’s action, the learner may not learn why its action was unsafe. Smooth composition is a natural fix.
2. The method is simple and practical. It does not require heavy online optimization and can reuse existing safe controllers.
3. The experiments cover both simple and more complex continuous-control domains, and the real-world CartPole result makes the paper more convincing than a simulation-only study.

**Weaknesses**
1. The safety guarantee depends heavily on the safe prior, the monitor, and the correctness of the recoverable set. The paper sometimes sounds more general than these assumptions justify.
2. Some experiments have high variance, and a few results are close to simpler baselines. The paper would be stronger with more ablations and more stress tests.

**Audience:**

Yes

**Audience Explanation:**

The paper is of interest to the safety community.

**Claims And Evidence:**

Yes

**Claims Explanation:**

Mostly yes. The main concern is that the broad “hard safety” claim depends heavily on assumptions: the safe prior must be valid, the safety monitor must be correct, the action space must allow meaningful convex interpolation, and the hard trigger must reliably recover safety at the boundary. The paper acknowledges some limitations, especially dependence on model-based safety design and local structured constraints, but the experiments do not fully stress-test these assumptions.

**Requested Changes:**

1. Clarify the exact safety guarantee. The paper should distinguish empirical zero violations, recoverability inherited from the safe prior, and any formal hard-safety guarantee. The current wording sometimes sounds stronger than what is proven. The authors should state precisely when convex composition preserves safety and when safety is only ensured by the final hard trigger.
2. Stress-test the safe prior and monitor assumptions. Since the method depends on a certified safe policy and safety margin, the paper should test inaccurate dynamics, noisy monitors, overly conservative priors, and suboptimal safe controllers.
3. Strengthen the comparison to closely related blending methods. AdaLam, Residual RL, and other policy-prior methods are close in spirit. The paper should more clearly isolate what is gained by the proposed risk-calibrated $\lambda$, variance damping, and hard endpoint trigger rather than by simply adding a safe prior.
4. Add stronger ablations. Please include fixed $\lambda$, fixed $p$, no variance damping, no learned sharpness, and hard-trigger-only variants. These would show whether the main improvement comes from smooth composition, the learnable sharpness, the certified prior, or tuning details.

---

### Review · Reviewer_9cDQ · 2026-07-13

**Summary Of Contributions:**

The authors propose a new method for online reinforcement learning under safety constraints. The main contribution over existing works is that it guarantees safety and smooth optimization dynamics. Smooth optimization dynamics is useful during training, and also ensures policies remain smooth, and hence are more interpretable.

Strengths:
- the method seems relatively straightforward to implement
- generally improves upon other methods in terms of training cost and number of safety violations

Weaknesses:
- I'm concerned about the applicability to high-dimensional setting (see below)
- The method seems to only be able to handle hard constraints (see below)

**Audience:**

Yes

**Audience Explanation:**

This paper seems to be relevant to the broader reinforcement learning community.

**Claims And Evidence:**

Yes

**Claims Explanation:**

Yes, the paper provides proofs for theoretical claims and thorough experiments with details.

**Requested Changes:**

[Q1] In high-dimensional space, $\Delta_{min}$ (for instance on the simplex example in Figure 3) needs to be $O(1/d)$ to ensure that $\Omega_{\Delta){min}}$ occupies a constant fraction of $S_C$. I suspect that in such settings, the method will essentially to default using $\pi^{safe}$ if $\Delta_{min}\ge \omega(1/d)$.
- Can the authors provide some discussion how to choose $\Delta_{min}$?
- Can the authors also provide some more high-dimensional experiments (say d=50 or 100)?
I'm not so familiar with this literature, so I'm not sure whether this is a primary limitation of this method, or of other methods as well.

[W2] Some safety constraints don't appear as just "hard constraints" as studied in the paper. For instance, we may have a daily safety constraint that allows unlimited decibels under 60 dB, 4 hours between 60-70 dB, 2 hours between 70-80, etc. Is it possible to handle these kind of constraints using AutoSafe?

I'm not super familiar with related literature, but [Q1] will be an important factor in my final decision. [Q2] would mainly strengthen the work.